# Porous isoreticular non-metal organic frameworks

Megan O'Shaughnessy[1], Joseph Glover[2], Roohollah Hafizi[2], Mounib Barhi[3], Rob Clowes[1], Samantha Y. Chong[1,4], Stephen P. Argent[5], Graeme M. Day[2✉] & Andrew I. Cooper[1,4✉]

Metal–organic frameworks (MOFs) are useful synthetic materials that are built by the programmed assembly of metal nodes and organic linkers[1]. The success of MOFs results from the isoreticular principle[2], which allows families of structurally analogous frameworks to be built in a predictable way. This relies on directional coordinate covalent bonding to define the framework geometry. However, isoreticular strategies do not translate to other common crystalline solids, such as organic salts[3–5], in which the intermolecular ionic bonding is less directional. Here we show that chemical knowledge can be combined with computational crystal-structure prediction[6] (CSP) to design porous organic ammonium halide salts that contain no metals. The nodes in these salt frameworks are tightly packed ionic clusters that direct the materials to crystallize in specific ways, as demonstrated by the presence of well-defined spikes of low-energy, low-density isoreticular structures on the predicted lattice energy landscapes[7,8]. These energy landscapes allow us to select combinations of cations and anions that will form thermodynamically stable, porous salt frameworks with channel sizes, functionalities and geometries that can be predicted a priori. Some of these porous salts adsorb molecular guests such as iodine in quantities that exceed those of most MOFs, and this could be useful for applications such as radio-iodine capture[9–12]. More generally, the synthesis of these salts is scalable, involving simple acid–base neutralization, and the strategy makes it possible to create a family of non-metal organic frameworks that combine high ionic charge density with permanent porosity.

Porous crystalline solids are interesting both for their fundamental chemistry and their potential in applications such as gas capture, catalysis and molecular separations. The ability to position chemical functionality with atomic precision in crystalline porous frameworks has created properties that do not exist in classical porous materials, such as activated carbons. Porous crystalline solids can be divided into two classes: extended, covalently bonded frameworks, such as MOFs[1,2] and covalent organic frameworks (COFs)[13,14]; and porous molecular crystals, such as hydrogen-bonded frameworks (HOFs)[7,8,15,16] and porous organic cages[17]. Porous bonded frameworks exploit strong, directional covalent or coordinate covalent bonding, which underpins the isoreticular principle[2], whereby series of structurally related frameworks can be synthesized. By contrast, porous molecular crystals involve weaker, non-covalent intermolecular interactions. They are therefore harder to design for a specific, programmed function and are less amenable to generalization.

Crystalline porous organic salts (CPOS)[5] are a subclass of porous molecular solids that are composed of acids and bases assembled through ionic interactions. Work on this began in the early 1990s[3,4], before the first porous MOFs were discovered[18,19]. However, whereas considerable progress has been made in creating MOFs, work on CPOS materials has not had the same success. MOFs have the advantages of reticular design, high levels of permanent porosity and, in some cases, good physicochemical stability. By contrast, although porous salts show promise for some applications[5,20,21], they lack many of the basic design principles that apply to isoreticular frameworks. For example, porous molecular salts can be subject to polymorphism[22] because the interactions between net charges in organic salts are less directional than is the case for coordinate covalent bonding in MOFs.

The potential for polymorphism in neutral HOFs has been tackled by using a priori CSP[7,8] to map the landscape of stable crystal packing modes and hence to predict the resulting physical properties. Work on MOFs has also seen recent developments in CSP[23,24] that could be used to anticipate likely stable structures for particular metal–linker combinations. The covalent bonding between metal nodes and organic linkers requires periodic density functional theory to adequately describe the relative energies of alternative structures, unlike organic molecular CSP in which intermolecular force fields can often capture the balance between the competing non-bonded interactions. This makes searching for structures much more expensive in MOF CSP, so these

[1]Materials Innovation Factory and Department of Chemistry, University of Liverpool, Liverpool, UK. [2]Computational System Chemistry, School of Chemistry, University of Southampton, Southampton, UK. [3]Albert Crewe Centre for Electron Microscopy, University of Liverpool, Liverpool, UK. [4]Leverhulme Research Centre for Functional Materials Design, University of Liverpool, Liverpool, UK. [5]School of Chemistry, University of Nottingham, Nottingham, UK. ✉e-mail: g.m.day@soton.ac.uk; aicooper@liverpool.ac.uk

studies have made heavy use of symmetry to guide the placement of MOF building blocks during random structure searching to reduce the computational expense.

CSP has not yet been applied to CPOS materials. Moreover, CSP has only rarely been applied to organic salts[6,25,26] because of the difficulty in modelling the range of interactions that govern their structure, the conformational flexibility in the building blocks, and the high dimensionality of the energy landscape that results from having multiple independent units in the crystallographic asymmetric unit.

Recent advances in the crystal engineering of porous organic salts and related systems have shown that some directionality can be forced in salts by using nonpolar steric hindrance around the charged sites[27], but those materials were not porous. A level of reticular chemistry was made possible by using carboxylic acids and amidines[28], but this strategy is less generalizable than for isoreticular MOFs. Again, the amidine salts were not activated successfully to yield porous structures. It has been shown that guanidinium organodisulfonates, despite being formally metastable with respect to dense packings, can retain microporosity for extended periods[29]. More broadly, a review of CPOS materials in 2020 concluded: "Most crystalline porous organic salts formed by noncovalent bonding remain unstable, leading to collapse of the framework after removing the guest molecules"[5]. Compared with MOFs, there are very few porous organic salts, and most of those reported are not actually permanently porous.

Although permanently porous organic salts have proved hard to design, they retain a conceptual and practical allure. For example, a wide range of salt-forming reactions exists as a toolkit for forming porous salts, making them potentially analogous to MOFs without the metals. Furthermore, one might expect to find unique physical properties in all-organic porous salt frameworks that have a high density of permanent charges lining the pores.

Ammonium halides are an archetypal class of organic salts that have been widely studied in pharmaceutical chemistry, making up a sizeable percentage of drug molecules. However, they are largely unexplored in the area of porous frameworks. Densely packed ammonium salts have been reported[30,31] and some have the ability to capture sulfate ions[32]. A different dense ammonium salt has been reported[33] that could efficiently catalyse the reduction of U(IV) to U(VI). And last year, porous ammonium halide salts were reported that could adsorb gases such as krypton and xenon[34], although those materials were synthesized without any computational structural design.

We show here that ammonium halide salts can form porous, thermodynamically stable frameworks that can be targeted by using a priori CSP. We also demonstrate that these porous salts can form predictable, isoreticular structure families, as is the case for MOFs[1,2,33] and COFs[35,36]. For example, we show that isoreticular forms persist across compound families if the length of the amine linkers is extended. These porous salts show robust, desolvatable porosity and exhibit useful properties such as high levels of iodine capture[9–12].

## Reticular design principle

Typically, MOFs consist of positively charged metal nodes connected by negatively charged organic linkers[18] (Fig. 1a) or neutral coordinating linkers with counter-anions to balance the charge[19]. An inverse approach is to construct frameworks in which negatively charged nodes are connected by positively charged linkers (Fig. 1a). Our basic design principle was to use rigid organic linkers that bear multiple amine groups (Fig. 1b). We imagined that crystals of the halide salts of these linkers would necessarily pack such that the cations and anions were in close proximity, and that this clustering of the salt functionality, coupled with the rigidity and length of the linkers, might lead to permanent porosity. Beyond this simple mental picture, however, it was impossible to anticipate the precise packing in such crystals because of the lack of strong directional intermolecular bonding in ammonium halide

salts. For this reason, we applied CSP to explore the likely low-energy packing modes for these salts before synthesis.

## CSP to guide synthesis

First we explored a tetrahedral amine linker, tetrakis-(4-aminophenyl) methane (TAPM) (Fig. 1b). The energy-structure landscapes derived from CSP calculations for the hypothetical chloride and bromide salts of TAPM are shown in Extended Data Fig. 1. In both cases, the lowest-energy structures were predicted to be dense and non-porous, indicating that TAPM was not a promising candidate for stable, permanently porous CPOS materials, at least with halide counterions, even though structurally analogous anionic tetrahedral sulfonates had been used previously to create porous salts[5]. To test this predicted outcome, single crystals of TAPM.X (X = Cl or Br) (**TAPM.X**) were grown by reacting TAPM with either HCl or HBr. By exploring a range of crystallization conditions, we found that two polymorphs, **TAPM.X/P1** and **TAPM.X/P2**, can be formed and that these polymorphs are isostructural for both halides. CSP finds structures with the experimental crystal packings observed in the two polymorphic forms for both TAPM salts (Extended Data Fig. 1c,d). As predicted, both **TAPM.X/P1** and **TAPM.X/P2** are dense and non-porous. **TAPM.X/P1** was identified as the lower-energy structure on both CSP landscapes (Extended Data Fig. 1a,b) and it was found to be the dominant form, produced under most crystallization conditions. Although these two salts did not yield porous frameworks, the results gave us confidence in the CSP methodology for these challenging systems, in which the calculations involved five independent structural units: the TAPM tetracation and four halide anions.

Next, we did CSP calculations for a wider range of amines and halides to search for candidate porous salt frameworks. The predicted structural landscapes for three promising candidate salts are shown in Fig. 2a–c, all of which involve trigonal, triamine linkers. For each of these salts, the lowest-energy predicted structures have pore channels that would be large enough to accommodate molecular guests. Moreover, the energy-density distributions of predicted crystal structures show pronounced 'spikes' containing multiple porous crystal packings that are broadly isostructural. Such features of CSP landscapes had previously been shown for neutral HOFs[7,8,37] to correspond to deep, isolated basins on the lattice energy surface[38] and this is most obvious in the energy-density distributions for **TT.Br** and **TTBT.Cl** (TT denotes 4,4′,4″-(1,3,5-triazine-2,4,6-triyl)tris[benzenamine] and TTBT denotes 4′,4‴,4‴‴-(1,3,5-triazine-2,4,6-triyl)tris[[1,1′-biphenyl]−4-amine]). For **TAPT.Cl** (TAPT denotes 1,3,5-tris(4-aminophenyl)benzene), the additional structural dimension of cation flexibility obscures similar spikes in the overall energy-density distribution (TAPT-1,2 and 4 in Fig. 2a); nevertheless, spikes formed by some cation conformers are expected to correspond to isolated, deep basins on the combined intramolecular–intermolecular energy surface. The prediction of global energy-minimum porous structures for these salts contrasts sharply with the porous metastable polymorphs that we predicted for neutral HOFs, which had lattice energies of up to 50 kJ mol$^{-1}$ above that of the predicted close-packed crystal structures[7,8,37]. This provides an insight that organic salts might be suitable for creating intrinsic porosity, which is important for applications because metastable crystals are subject to porosity loss by densification[29].

The CSP calculations indicated that this series of salts might also be isoreticular, as is the case for MOFs[2]. That is, the linker with the longest arms, TTBT, was predicted to yield salts with crystal packings that are isoreticular with salts of the short-armed linkers, TT and TAPT, but with larger pore channels and a higher pore volume (Fig. 2e,f).

## Synthesis of porous salts

Bulk crystalline powders of **TAPT.Cl, TT.Br** and **TTBT.Cl** were isolated through simple dropwise addition of either HCl or HBr solutions into

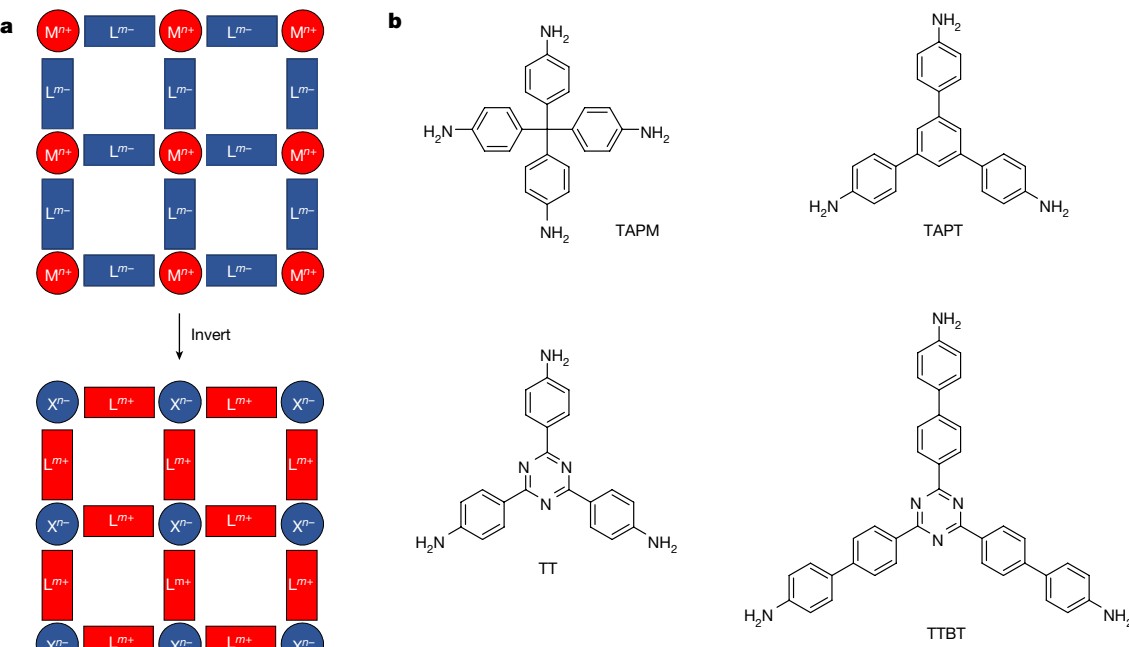

**Fig. 1 | Inverse reticular design strategy for porous salt frameworks. a**, Most MOFs (top) comprise positively charged metal nodes ($M^{n+}$) and negatively charged organic ligands ($L^{m-}$). An inverse strategy (bottom) is to design organic salt frameworks with negatively charged nodes ($X^{n-}$) and positively charged organic linkers ($L^{m+}$). **b**, Structures of the aniline derivatives used as linkers. These linkers take the form of cationic ammonium halide salts in the resulting salt frameworks.

solutions of the respective amines, whereby the salts precipitated instantaneously. Structural matches for the three isolated salts could be found on their CSP landscapes by comparing the predicted and experimental powder X-ray diffraction (PXRD) data patterns (Fig. 2g–i), as indicated by the red stars in Fig. 2a–c. In all three cases, these matches were found to lie at the tip of a spike in the CSP landscape, and for **TT.Br** and **TTBT.Cl**, these structures corresponded to the predicted global energy-minimum structures.

The three predicted crystal packings that best matched the experimental data were all isoreticular and comprised two distinct one-dimensional (1D) pore channels (Fig. 2d–f). The first pore (A) is defined by clusters or 'tubes' of the protonated amines and the halide counterions; it is cylindrical, highly charged and has a narrow pore diameter (4.86–5.87 Å). The second pore (B) is diamond shaped, is less polar and is defined by the aromatic linkers; this pore diameter is larger in **TTBT.Cl** (14.3 Å × 8.5 Å) than in **TAPT.Cl** (7.9 Å × 4.6 Å) or **TT. Br** (7.0 Å × 5.8 Å), but the dimensions of the ionic pore (A) are almost the same in all three predicted structures. This dual-channel structure leads to predicted pore volumes in these trigonal amine salts that are higher than for 4,4′,4′′,4′′′-(ethene-1,1,2,2-tetrayl)tetraaniline (ETTA) salts[34]. For example, using a probe radius of 1.2 Å, the calculated solvent-accessible pore volume for **TAPT.Cl** is 31.9% of the unit-cell volume, 31.4% for **TT.Br**, and 43.2% for the larger-pore isoreticular framework, **TTBT.Cl**. The equivalent calculated pore volume calculated for the chloride salt of ETTA (ETTA_Cl) is 18.2% (ref. 34). This four-arm linker, ETTA, is closer in structure and geometry to TAPM (Fig. 1b), which gives dense, non-porous salts here (Extended Data Fig. 1).

Single crystals of **TAPT.Cl** grown from methanol and chlorobenzene revealed that the salt crystallized in a trigonal space group, $P3m1$ (Supplementary Fig. 6). This experimental crystal structure, albeit for a partial solvate (**TAPT.Cl**, 1.25[$C_6H_5Cl$], 1.5[$H_2O$]), confirmed the match for the computationally predicted structure. In this solvate, the solvent occupies around 78% of the void volume. A structural overlay of the predicted and experimental structure is shown in Extended Data Fig. 2b.

This is, to our knowledge, the first experimental evidence that organic salts can form porous frameworks that follow an a priori atomistic structure prediction.

Crystals also formed for both **TT.Br** and **TTBT.Cl**, but the lower solubility and higher reactivity of those systems meant that publishable single-crystal X-ray data could not be obtained. Conversely, this high reactivity and low solubility allowed us to obtain multiple grams of (poly)crystalline powders for these materials within hours by the simple dropwise addition of acid at room temperature (Methods). Comparison of experimental PXRD data with equivalent data derived from the global energy-minimum CSP structures (Fig. 2h,i) suggested that **TT.Br** and **TTBT.Cl** formed crystal packings that were broadly isoreticular; that is, extension of the organic linkers led to larger pores, as for isoreticular MOFs. Given the challenges in growing single crystals, high-resolution transmission electron microscopy (HR-TEM) was used to further characterize these materials, as used to provide structural information for other porous frameworks[39,40]. HR-TEM further demonstrates the crystallinity of **TT.Br** and **TTBT.Cl** (Extended Data Fig. 3a,b). The highly crystalline structure of **TT.Br** is shown in Extended Data Fig. 3a; the inset shows an expansion of the rectangle in the middle of the particle, showing the anticipated pore structure. To further confirm the proposed crystal packing of **TT.Br**, we compared the fast Fourier transform (FFT) of the experimental HR-TEM (Extended Data Fig. 3a, top right) with the simulated electron diffraction pattern for the best-matched CSP structure (Extended Data Fig. 2a, bottom right). The yellow and blue dashed circles in both patterns correspond to the first and second hexagonal order, with d-spacing values of 1.572 nm and 0.92 nm, respectively. For **TTBT.Cl** (Extended Data Fig. 3b), HR-TEM shows the 1D pore channels in the crystal and the corresponding FFT image (Extended Data Fig. 3b, top right) shows the (010), (020) and (030) axes with an alignment along the [010] zone axis determined after comparison with the simulated pattern from the best-matched CSP-derived structure (Extended Data Fig. 3b, bottom right). The halos observed in the FFT images indicate that there is uncorrelated disorder in these two materials, particularly for **TTBT.Cl** (Extended Data Fig. 3b). The crystal structure of **TTBT.Cl**

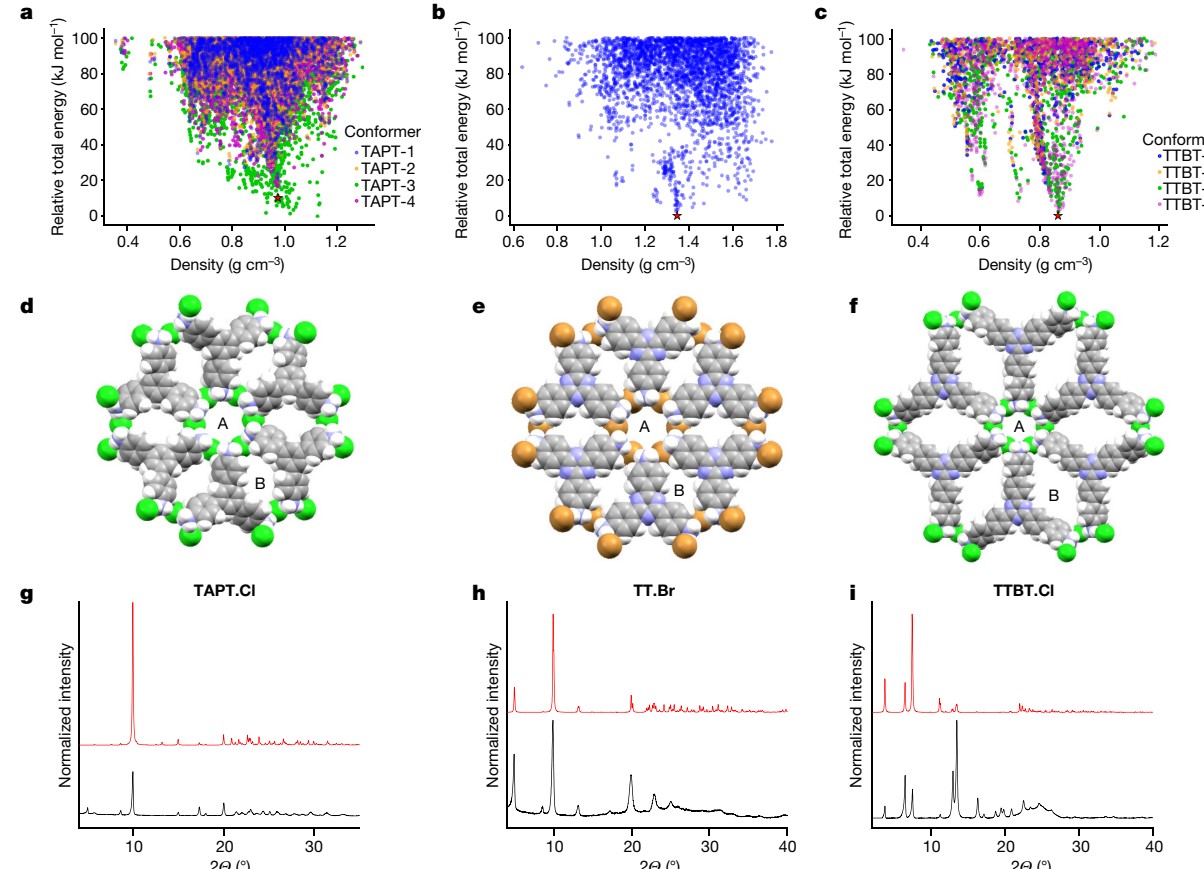

**Fig. 2 | CSP suggests porous, isoreticular ammonium halide salts. a–c**, CSP energy landscapes for **TAPT.Cl** (**a**), **TT.Br** (**b**) and **TTBT.Cl** (**c**). For TAPT and TTBT, we considered the four lowest-energy molecular conformers predicted for the amine linker, and these are colour-coded in the CSP landscapes. TT was assumed to adopt a planar conformation (the computed minimum-energy conformer) because of its triazine core. We carried out CSP for both the chloride and bromide salts for all three systems; the **TT.Br** landscape is shown because this salt crystallized more effectively than did the corresponding chloride. For all three salts, the lowest-energy, most thermodynamically stable structures are predicted to be porous. **d–f**, Space-filling representations of the closest

CSP matches (indicated by stars in **a**–**c**) found for the observed experimental structures of **TAPT.Cl** (**d**), **TT.Br** (**e**) and **TTBT.Cl** (**f**). For **TT.Br** and **TTBT.Cl**, the closest matches are the global minimum-energy structures. The labels A and B denote the two kinds of pores in these three isoreticular frameworks. **g–i**, Comparison of the experimental PXRD data (bottom, black) with the PXRD pattern predicted from the closest match in the CSP landscape. 2$\theta$ is the diffraction angle, which is the angle between the incident and the detected X-rays. (top, red) for **TAPT.Cl** (**g**), **TT.Br** (**h**) and **TTBT.Cl** (**i**). Details of the analyses for **TAPT.Cl** and **TT.Br**, and the refinement of **TTBT.Cl**, are given in Supplementary Information section 2.

was solved using PXRD data (Extended Data Fig. 3c). All these data support the structure assignments made in Fig. 2.

## Ionic interactions create porosity

Analysis of the CSP energy landscapes reveals that our initial design hypothesis of charge adjacency was satisfied, at least for the three trigonal amine linkers. The anion coordination of the ammonium groups is summarized in Fig. 3, represented as the mean count of nitrogen–halide close contacts per ammonium in each predicted crystal structure. Low-density spikes of structures correspond to the maximization of such close contacts, reaching an average of four close halides per ammonium in the most-stable porous structures. That is, porosity arises from the crystal packing constraints introduced by closely adjacent net charges attached to rigid aromatic linkers. The trigonal organic linkers express this charge adjacency more effectively than the two tetrahedral salts do. For **TAPM.Cl** and **TAPM.Br**, there are no low-energy crystal structures predicted that allow more than three close contacts between the ammonium nitrogen atoms in the organic linkers and the halide nodes (Fig. 3a,b). This means that the electrostatic interactions do not overrule the stabilization that can be achieved by close packing. Therefore, these salts do not form open porous frameworks.

By contrast, the trigonal amine linkers **TT.Br** and **TTBT.Cl** yield CSP landscapes that contain structures with four close halide contacts per ammonium nitrogen (Fig. 3d,e); these structures predominate in the spikes of low-energy, low-density predicted packings. The **TT.Br** and **TTBT.Cl** energy landscapes show clearly why an open, low-density structure is formed: structures with higher densities form fewer ammonium–halide close contacts and are therefore less energetically stable. For **TAPT.Cl**, the predicted crystal packing that corresponds to the experiment has three close chloride contacts per ammonium, according to the distance thresholds in Fig. 3. However, the number of close contacts is sensitive to the threshold value, and for **TAPT.Cl** it increases to four chloride close contacts around two arms of the cation, with a 25% increase in the distance cutoff (Supplementary Fig. 34). For all three trigonal amine salts (Fig. 2d–f), the packing arrangement with four close halide–ammonium contacts is expressed by the cylindrical ionic pores (A) (Fig. 3f), which in turn create the diamond-shaped pores (B).

## Other potential porous polymorphs

As found for neutral HOFs[7,8,37], these CSP landscapes suggest that other porous polymorphs might also be accessible in the laboratory (Extended Data Figs. 4–6). For example, alternative porous structures

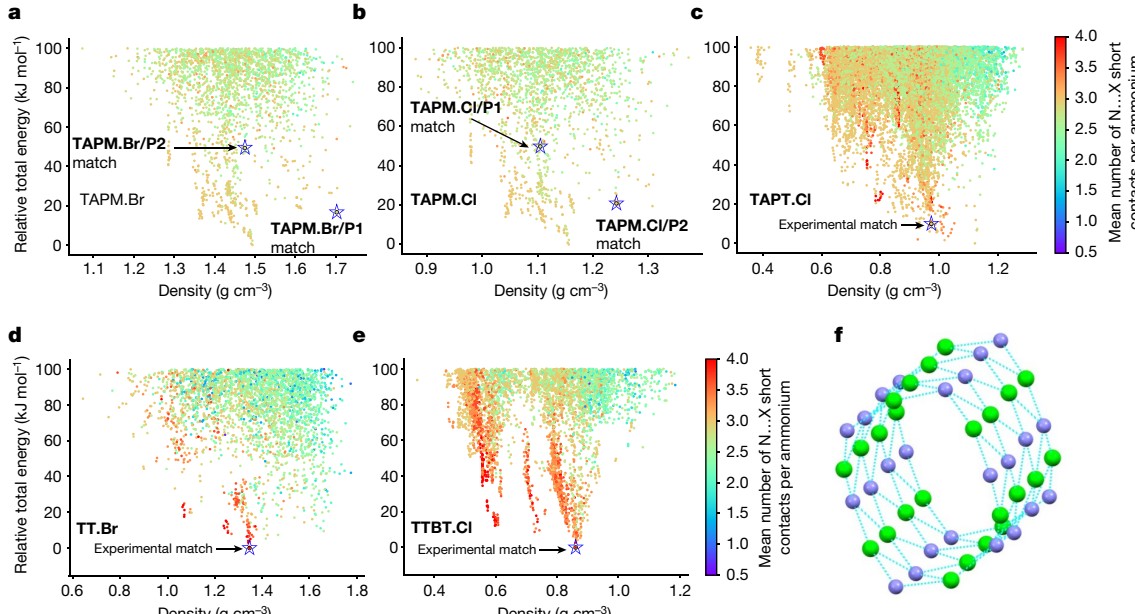

**Fig. 3 | Charge adjacency dictates crystal packing. a–e**, CSP plots colour coded by the mean number of short contacts (below the sum of nitrogen and halide van der Waals radii, which are 3.3 Å for chloride salts and 3.4 Å for bromide salts) between ammonium nitrogens and halide ions for **TAPM.Br** (**a**), **TAPM.Cl** (**b**), **TAPT.Cl** (**c**), **TT.Br** (**d**) and **TTBT.Cl** (**e**). Experimentally observed dominant polymorphs are indicated by a star. **f**, Global energy-minimum predicted structure for **TTBT.Cl** emphasizing the four short ammonium–halide contacts (blue dotted lines) that define the cylindrical pore channel (A) in Fig. 2f. Purple, nitrogen atoms in the $NH_3^+$ cation (hydrogens not shown); green, $Cl^-$ anions.

were predicted for **TAPT.Cl**, some of which had lower predicted lattice energies than the experimentally observed crystal packing (such as **TAPT.Cl**/4 and **TAPT.Cl**/5 in Extended Data Fig. 4), at least in the absence of solvent. We also predicted structures with much higher pore volumes, such as **TAPT.Cl**/3 (Extended Data Fig. 4), which lies 29.7 kJ $mol^{-1}$ above the global minimum. By analogy with HOFs, in which solvent stabilization of at least 50 kJ $mol^{-1}$ has been observed for porous polymorphs[7,8,37], **TAPT.Cl**/3 or a nearby structure in the predicted set could in principle be accessible under different experimental conditions. Likewise, the CSP landscapes for **TT.Br** (Extended Data Fig. 5) and **TTBT. Cl** (Extended Data Fig. 6) indicate that there might be other potential porous forms in a relative lattice energy window of 50 kJ $mol^{-1}$, although in these two cases the experimentally observed polymorphs match the global minimum-energy predicted structures (Fig. 2b,c). Closely related crystalline forms can be found across the three energy landscapes. For example, **TT.Br**/4 and **TTBT.Cl**/3 are isoreticular with **TAPT.Cl**/3 (Extended Data Fig. 7). Similarly, **TAPT.Cl**/2, **TT.Br**/1 and **TTBT.Cl**/1 are all isoreticular (albeit with high relative lattice energies), as are the structures of **TT.Br**/5 and **TTBT.Cl**/4, and the structures of **TAPT.Cl**/4 and **TT.Br**/6, which have more competitive predicted lattice energies.

We can also use energy–structure–function maps[7] to predict global property tendencies. For example, the dominance of red data points in the energy–structure–function maps coloured by pore channel dimensionality (Extended Data Fig. 8a–c) shows that all three of these salts would be expected to express 1D channel geometries, rather than 2D or 3D channels, if a porous structure is formed. Likewise, energy–structure–function maps suggest that **TTBT.Cl** has the greatest chance of forming a mesoporous structure with pores larger than 2 nm in diameter (Extended Data Fig. 8d–f), such as **TTBT.Cl**/3, although only microporous materials (with pores narrower than 2 nm) were observed under the experimental conditions tested here.

### Gas sorption in porous salts

These materials did not adsorb much nitrogen at 77 K, but this is relatively common for small-pore microporous solids and may reflect limited guest mobility at such temperatures. By contrast, all three porous salts adsorbed carbon dioxide reversibly at temperatures in the range 195 K to 298 K (Extended Data Fig. 9). **TAPT.Cl** and **TT.Br** adsorb 4.0 mmol $g^{-1}$ and 4.6 mmol $g^{-1}$ $CO_2$, respectively, at 195 K as saturation is approached. **TT.Br** in particular adsorbs more $CO_2$ than do the other porous organic salts at this temperature[5,34,41,42]. By contrast, **TTBT.Cl** adsorbed around 2.4 mmol $g^{-1}$ despite having the lowest predicted crystal density and the largest nominal pore volume (Fig. 2). The $CO_2$ desorption isotherm for **TTBT.Cl** at this temperature also shows pronounced hysteresis, unlike those of **TAPT.Cl** and **TT.Br**, and the isotherm collection time was long (more than 90 h). It is likely that the low-density **TTBT.Cl** crystals are less stable under the degassing conditions used here (14 h at either 80 °C or 110 °C), so some porosity is lost during sample preparation or during sorption and desorption (Supplementary Fig. 14). By contrast, as discussed below, **TTBT.Cl** exhibits the highest capacity, good stability and the most rapid adsorption kinetics for guests such as iodine when activated under less-rigorous conditions. Unlike the $CO_2$ isotherms, this supports our prediction that **TTBT.Cl** should have the greatest pore volume in this isoreticular series of porous salts.

### Porous salts for iodine capture

The capture of radio-iodine is important in the nuclear industry and for environmental protection[9–12,43]. These salts, which have highly polar pore channels, struck us as potentially useful adsorbents for iodine capture. Non-porous **TAPM.Cl**/P1 showed little iodine uptake (3.6 wt%; Fig. 4a), which we ascribed to adsorption on the crystal surface. By contrast, the three porous salts, **TAPT.Cl**, **TT.Br** and **TTBT.Cl**, showed high iodine uptakes of 248 wt% (6.8 mol per mol), 213 wt% (4.99 mol per mol) and 211 wt% (3.83 mol per mol), respectively. These iodine uptakes outperform most MOFs studied at comparable temperatures (60–80 °C), including ZIF-8 (125 wt% iodine)[9], Cu-BTC (175 wt%)[10], MFM-300(Sc) (154 wt%)[11] and NU-1000 (145 wt%)[12]. Indeed, a review[43] suggests that **TTBT.Cl** adsorbs more iodine than all but five MOFs reported, all of which have high surface areas (more than 2,000 $m^2$ $g^{-1}$) and lower

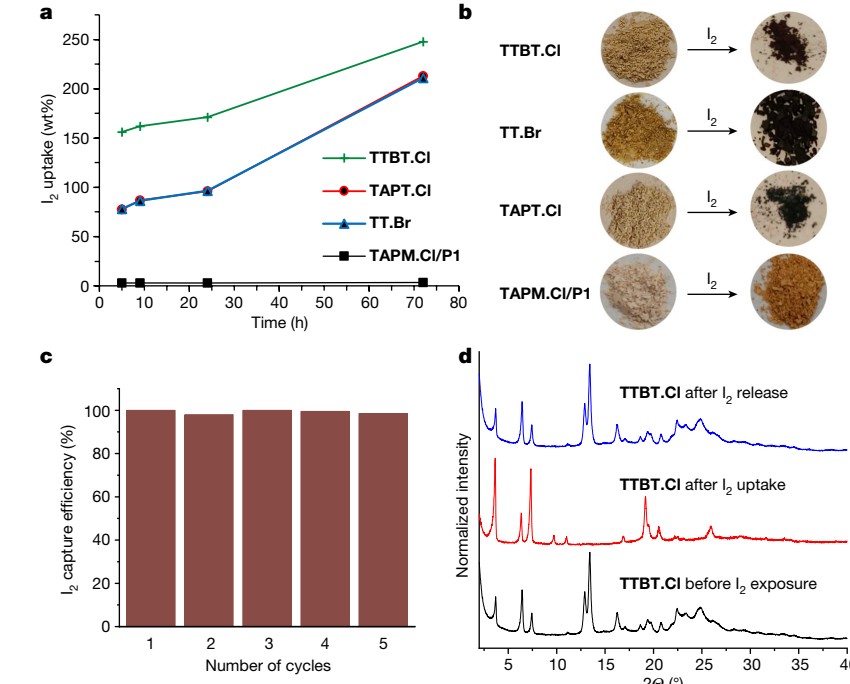

**Fig. 4 | Reversible iodine uptake in porous organic salts. a**, Iodine uptake in porous **TAPT.Cl**, **TT.Br** and **TTBT.Cl** frameworks and non-porous **TAPM.Cl/P1** as a function of time. Note that the **TAPT.Cl** and **TT.Br** plots overlay almost exactly. **b**, Photographs of the salts before (left) and after (right) exposure to iodine. **c**, Recyclability tests over five cycles for **TTBT.Cl**; 100% efficiency is defined as the initial iodide uptake (cycle 1). **d**, PXRD data showing the reversibility of iodine adsorption in **TTBT.Cl**; similar reversibility was observed for **TAPT.Cl** and **TT.Br** (Supplementary Figs. 20 and 21).

framework densities than this non-metal organic framework. Moreover, the iodine uptake in these salt frameworks is reversible over multiple cycles (Fig. 4c,d).

## Water stability

The stability of frameworks to water is another important practical consideration. For these ammonium halide salts, this depends on the organic linker. **TAPT.Cl** is water soluble, whereas **TT.Br** has very low water solubility but becomes amorphous when immersed in water. By contrast, the more-hydrophobic **TTBT.Cl** framework is insoluble in water and a sample submerged in water was shown by PXRD to have stable crystallinity for at least 48 h (Supplementary Fig. 23). Water adsorption isotherms were also collected for **TTBT.Cl** and it was shown to adsorb 12.4 mmol g$^{-1}$ water (Supplementary Fig. 24). PXRD analysis before and after water sorption showed that the sample had retained a good level of crystallinity.

## Outlook

In this study, we have introduced a computational design-led strategy for non-metal-containing framework materials. The frameworks can be produced on multigram scales from abundant elements by the simple dropwise addition of acid to solutions of the amine linkers. The first examples of these materials already show practical promise, outperforming most MOFs for iodine capture[43]. Other applications might take advantage of the highly charged pore channels (Figs. 2 and 3f), such as proton conduction, catalysis, water capture or hydrogen storage.

These frameworks can be thought of as 'inverted' MOFs in which the halide anions are analogous to the metal cations (Fig. 1a); that is, non-metal organic frameworks. Just as MOFs can be structurally diversified by changing the metal nodes and the organic linkers, it should be possible to create similar families of non-metal organic frameworks. We chose ammonium halides here because they are easy to synthesize and

are well known in a pharmaceutical context, but this inverse reticular strategy should be diversifiable. For example, a wide variety of other counter-ions can be considered, such as nitrates, sulfates and hydrogen sulfates, tetrafluoroborates, hydrogen carbonates, phosphates, cyclic phosphates, arsenates, carboxylates and tetrafluoroborates, all of which are known to form salts with ammonium cations. Mixed-anion frameworks are also possible[34], although predicting the most stable salt composition a priori could be computationally expensive. Just as for MOFs, a range of organic amine linkers can be conceived, including aliphatic amines, if they are sufficiently rigid, as well as pyridinium or imidazolium analogues.

Like MOFs, but unlike other covalent non-metal frameworks, such as COFs[13,14], these materials are synthesized by salt formation. This is reversible enough to produce single crystalline materials (Supplementary Fig. 6), which is still relatively uncommon for COFs[44–48]. These molecular salts also have properties that are not found in bonded frameworks, such as solubility in certain solvents (Supplementary Figs. 5, 7 and 10), which may aid processing and purification.

We observed polymorphism for **TAPM.X** salts (Extended Data Fig. 1), and CSP calculations suggest that other ammonium halide salts might in principle also be polymorphic (Extended Data Figs. 4–6), although CSP is known to overpredict polymorphism[49,50]. Interestingly, polymorphism has not been observed in crystalline guanidinium organosulfonate materials[3,4,29]. This might be due to more directional hydrogen bonding between guanidinium ions and sulfonate groups restricting the possibilities for low-energy crystal packings, in comparison with ammonium halide salts that comprise simple spherical anions. This could have broader implications for the design of non-metal organic frameworks using anions other than halides.

We see CSP as the key to exploring this area because the ionic bonding in these salts is weaker and less directional than for most MOFs, and CSP allows us to evaluate the propensity for new combinations of organic cations and counterions to form stable, porous crystals before synthesis. This will identify candidates for porous frameworks from the

much larger pool of organic salts that form dense, non-porous crystals, or that cannot be activated because they are metastable[5]. Moreover, the ability to predict frameworks that have thermodynamically stable porous forms is a major advantage for finding applications. The robust porosity of **TTBT.Cl** and its stability to multiple iodine sorption–desorption cycles (Fig. 4c) can be explained by the absence of denser, more-stable packings available to this crystal (Fig. 2c). This is not the case for most hydrogen-bonded organic frameworks, nor indeed for many MOFs. For example, our earlier neutral porous HOFs[7] may be unsuitable for iodine-capture applications because there are multiple, denser polymorphs available that are more stable, and the porosity would be lost under practical capture conditions.

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

# Methods

## CSP

Two of the four amines studied, TAPM and TT, have only one predicted conformer (Supplementary Fig. 25). By contrast, TAPT and TTBT both have four predicted low-energy conformers (Supplementary Figs. 26 and 27). These molecular structures were obtained by performing a conformer search using the low-mode sampling method in the Schrödinger MacroModel software package, with energies being modelled using the OPLS2005 force field[51,52]. For each amine, the unique conformers from the search were reoptimized using density functional theory at the PBE0/6-311 G** level of theory with GD3BJ empirical dispersion correction, as implemented in the Gaussian09 software package[53,54]. The TT, TAPT and TTBT cations were assigned a charge of +3, whereas the TAPM cation was assigned a charge of +4. Owing to the small energy difference between conformers (a spread of less than a 1.5 kJ mol$^{-1}$), they have the same chance of forming a low-energy crystal structure, so each of these conformers was used as a starting point for CSP calculations.

CSP was done using the Global Lattice Energy Explorer program[55], which uses low-discrepancy, quasi-random sampling of crystal packing variables to produce uniform sampling of the lattice energy surface. The cation geometry was kept rigid throughout the initial CSP process. Trial crystal structures were generated across 11 space groups and their lattice energies were minimized until a target number of valid crystal structures was met (Supplementary Table 8). Crystal structures were generated with one cation in the asymmetric unit cell and $X$ anions, where $X = 3$ for TT, TAPT and TTBT, and $X = 4$ for TAPM. Rigid-molecule lattice energy optimizations were performed using the DMACRYS software[56]. Lattice energies were calculated using an anisotropic atom–atom energy model based on a revised version of the Williams 99 force-field, combined with atom-centred multipoles calculated from a distributed multipole analysis of the PBE0/6-311 G** density[57,58]. Multipoles up to hexadecapole on each atom were included, and the polarizable continuum model was applied to the distributed multipole analysis to further improve the electrostatic model using a dielectric constant of 3.0. Chloride parameters were taken from a previously published fitting[59], and bromide parameters were taken from molecular-dynamics studies of ionic liquids[60]. The bromide parameters were deemed suitable from the results of a small CSP test on bromide salts in the Cambridge Structural Database (Supplementary Figs. 28 and 29). Duplicate crystal structures were removed from the final CSP landscape by calculating the similarities of simulated PXRD patterns.

To test the sensitivity of the CSP to the final energy model and the rigid-cation approximation during lattice energy minimization, predicted crystal structures of **TAPM.Cl** were reoptimized using third-order density functional tight-binding theory with self-consistent charges, as implemented in DFTB+[61,62]. We used 3OB Slater–Koster parameter files for all simulations, and hydrogen-containing pair-potentials were further damped using an exponent of 4.0. We corrected for missing London dispersion interactions with the atom-pairwise D3 scheme in the rational damping variant[52]. The tolerance for self-consistent charges was set to $10^{-5}$ atomic units and each optimization was considered converged when the maximum force on each atom dropped below 0.03 eV per Å. The k-point grid sampling was set to achieve a maximum spacing of 0.05 Å$^{-1}$ between sampled k-points. Dispersion interactions were accounted for using the D3 correction with the following parameters: $S_6 = 1.0$, $S_8 = 0.0$, $a_1 = 0.841$ and $a_2 = 3.834$, in a cutoff radius of 64 Å, where $S$ and $a$ are adjustable parameters that scale the damping term of the dispersion energy. These parameters have previously been shown to improve the geometrical properties of large molecular crystals when optimized[62]. Crystal structures that were deemed a match with experimental crystal structures were further optimized using periodic density functional theory, as implemented in the VASP software package[63–67]. VASP calculations involved two steps with the first step involving optimization of both atomic positions and unit-cell parameters, and the second step involved a single-point energy calculation. All optimizations used the PBE exchange correlation functional with Grimmes D3(BJ) dispersion correction[54]. The projector-augmented-wave method was used for all calculations with the standard supplied pseudopotentials. A plane wave cutoff of 500 eV was used in all calculations and k-point sampling was done using a regular k-point mesh with a maximum spacing of 0.05 Å$^{-1}$ between sampled k-points. Optimizations were considered converged when the forces on all atoms were smaller than 0.03 eV per Å.

## Synthesis of porous salts

The organic amine linkers were synthesized using previous methods[68]. The halide salts were synthesized by simple dropwise addition of HCl or HBr solutions to the amine linkers and can be prepared on multigram scales. The specific conditions for each salt are provided in Supplementary Information section 2. As an example, bulk powders of **TAPT.Cl** were formed by dissolving 2 g TAPT in a good solvent (such as tetrahydrofuran; see Supplementary Table 2 for a list of good and bad solvents) at a concentration of 5 mg ml$^{-1}$. Methanolic HCl was then added dropwise over 5 min with stirring at room temperature. The mixture was stirred for a further 1 h before the solvent was removed under reduced pressure, and the resulting solid was washed with tetrahydrofuran, providing crystalline **TAPT.Cl** as an off-white solid (2.57 g, 98% yield).

## Single crystal growth

Specific crystallization conditions are given in the Supplementary Information section 2. As an example, single crystals of **TAPT.Cl** suitable for X-ray diffraction were grown by dissolving 3 mg **TAPT.Cl** in a mixture of methanol (0.5 ml) and chlorobenzene (0.05 ml). The solvent was then left to evaporate at room temperature for 16 h, resulting in block crystals of **TAPT.Cl**.

## Crystallography

PXRD data were collected in transmission mode on powder samples held on thin Mylar films in aluminium well plates using a Panalytical Empyrean diffractometer equipped with a high-throughput screening XYZ stage, X-ray focusing mirror and a PIXcel detector using Cu-Kα (wavelength $\lambda = 1.541$ Å) radiation. PXRD data for further structural analysis of **TTBT.Cl** were collected using a sample contained in a 0.7 mm borosilicate glass capillary tube on the same instrument in Debye-Scherrer geometry with a capillary spinner. Data were collected over the diffraction angle range $2° \le 2\theta \le 50°$ with a step size of 0.013° over 4 h, and the data were analysed using TOPAS Academic[69] for indexing, structure solution and refinement. Single-crystal X-ray data were obtained using a Rigaku MicroMax-007 HF rotating anode diffractometer (Mo-Kα radiation, $\lambda = 0.71073$ Å, Kappa 4-circle goniometer, Rigaku Saturn724+ detector), or at beamline I19 at the Diamond Light Source in Didcot, UK, using silicon double-crystal monochromated synchrotron radiation ($\lambda = 0.6889$ Å, Pilatus 2M detector). Data reduction was performed using CrysAlisPro. Structures were solved with SHELXT[70] and refined by full-matrix least-squares on $|F|^2$ by SHELXL[71], interfaced through the program OLEX2 (ref. 72). All non-hydrogen atoms were refined anisotropically, and all hydrogen atoms were fixed in geometrically estimated positions and refined using the riding model. For full refinement details, see Supplementary Tables 3 and 4.

## HR-TEM

The salt samples were dispersed in chloroform at a concentration of 1 mg ml$^{-1}$ using sonication. The samples were then dropcast onto copper grids. TEM images were obtained using a JEOL 2100+ microscope operating at 200 kV and equipped with a Gatan Rio Camera.

## Gas sorption analysis

Nitrogen isotherms were collected at 77 K using a Micromeritics ASAP2420 volumetric adsorption analyser. Carbon dioxide isotherms

were collected up to a pressure of 1,200 mbar on a Micrometrics ASAP2020 volumetric adsorption analyser at 273 K or 298 K. Carbon dioxide isotherms at 195 K were collected using Micromeritics 3flex volumetric adsorption analyser. All samples were activated at 110 °C for 14 h under vacuum for all gas sorptions except for the carbon dioxide isotherms measured at 195 K, for which the samples were activated at 353 K for 14 h under vacuum.

## Iodine-capture experiments

Iodine adsorption tests were done on five separate samples of each porous salt at 70 °C to volatilize the iodine; the adsorption results were averaged across the five samples. The salt samples were heated under vacuum at 343 K for 16 h to remove the organic solvent before the iodine-capture experiments were done. To perform the tests, the salts were placed into 4-ml vials, which were then placed into larger 14-ml vials containing an excess of iodine. The outer vial was then sealed and placed into an incubator at 343 K. At particular time intervals, the samples were removed from the incubator and, once cooled to room temperature, the inner vial containing the salts was removed and the vial was weighed to monitor the iodine uptake. For each salt, five samples were tested (5 mg salt). All five samples gave consistent uptakes and the average uptake values were used. To test the recyclability of the salts for iodine capture, the iodine was removed and the capture was repeated over five cycles. Owing to the high mass of iodine captured in each system, and the slow release using the vacuum alone, chloroform was first used to remove the bulk of the iodine before the final traces were removed at 70 °C under vacuum for 16 h (Supplementary Fig. 19). Thermogravimetric analysis and PXRD further confirmed the capture of iodine in these three frameworks (Supplementary Figs. 15–22). To test the recyclability of these salts, the iodine was removed after adsorption by extraction with chloroform, followed by evacuation. Sample colour, thermogravimetric analysis and PXRD data confirmed that the iodine was eventually removed in all three systems. The samples were subjected to five cycles of iodine capture and release, demonstrating excellent recyclability for all three porous salts (Fig. 4c and Supplementary Figs. 20 and 21). PXRD data indicated that a structural change occurs in all three porous salts at the point of iodine adsorption, as might be expected for such high guest uptakes, but the original guest-free crystal structure was regenerated for all three frameworks after iodine removal (Fig. 4d).

## Data availability

The crystal-structure data are in the Supplementary Information. CSP structures with calculated energies and properties are available at https://doi.org/10.5258/SOTON/D2857. The experimentally determined crystal structures, including structure factors, have been deposited as CIFs with the Cambridge Crystallographic Data Centre (entries 2308598 (**TAPM.Cl/P1**), 2308599 (**TAPM.Cl/P2**), 2308596 (**TAPM.Br/P1**), 2308597 (**TAPM.Br/P2**), 2308662 (**TAPT.Cl**)). CIFs are available free of charge at http://www.ccdc.cam.ac.uk/data_request/cif.

## Code availability

The code implementing the CSP methods is being prepared for open source release in early 2024. In advance of release, please contact G.M.D. at g.m.day@soton.ac.uk.

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

**Acknowledgements** This project has received funding from the European Research Council under the European Union's Horizon 2020 research and innovation program (grant agreement 856405). The authors also received funding from the Engineering and Physical Sciences Research Council (EPSRC, EP/V026887/1) and the Leverhulme Trust via the Leverhulme Research Centre for Functional Materials Design. A.I.C. thanks the Royal Society for a Research Professorship (RSRP\S2\232003). Through our membership of the UK HEC Materials Chemistry Consortium, which is funded by EPSRC (EP/L000202), this work used the computational resources of the UK Materials and Molecular Modelling Hub, which is partly funded by EPSRC (EP/T022213/1, EP/W032260/1 and EP/P020194/1). The authors acknowledge the IRIDIS High Performance Computing Facility and associated support services at the University of Southampton in the completion of this work. The authors thank K. Arnold and O. Gallagher for collecting scanning electron microscopy images. We acknowledge the Diamond Light Source for access to beamline I19. M.O'S. thanks B. Li, A. He and A. Kai for support during the synchrotron diffraction experiments.

**Author contributions** M.O'S. led the experimental work. M.O'S. and R.C. collected the gas sorption data. M.O'S. and S.P.A. solved and refined the **TAPT.Cl** crystal structure. M.O'S. did all the other experimental work, including synthesis, crystallizations, single-crystal X-ray diffraction and PXRD, and the iodine-uptake experiments and analysis. J.G. and R.H. performed and analysed the CSP calculations. G.M.D. led the computational work. M.B. obtained and interpreted the HR-TEM data. S.Y.C. carried out PXRD analyses to solve the **TTBT.Cl** structure. A.I.C. conceived the idea, supervised the experimental work and carried out the initial CSP calculations. All authors contributed to the preparation of the manuscript.

**Competing interests** The authors declare no competing interests.

**Additional information**
**Correspondence and requests for materials** should be addressed to Graeme M. Day or Andrew I. Cooper.

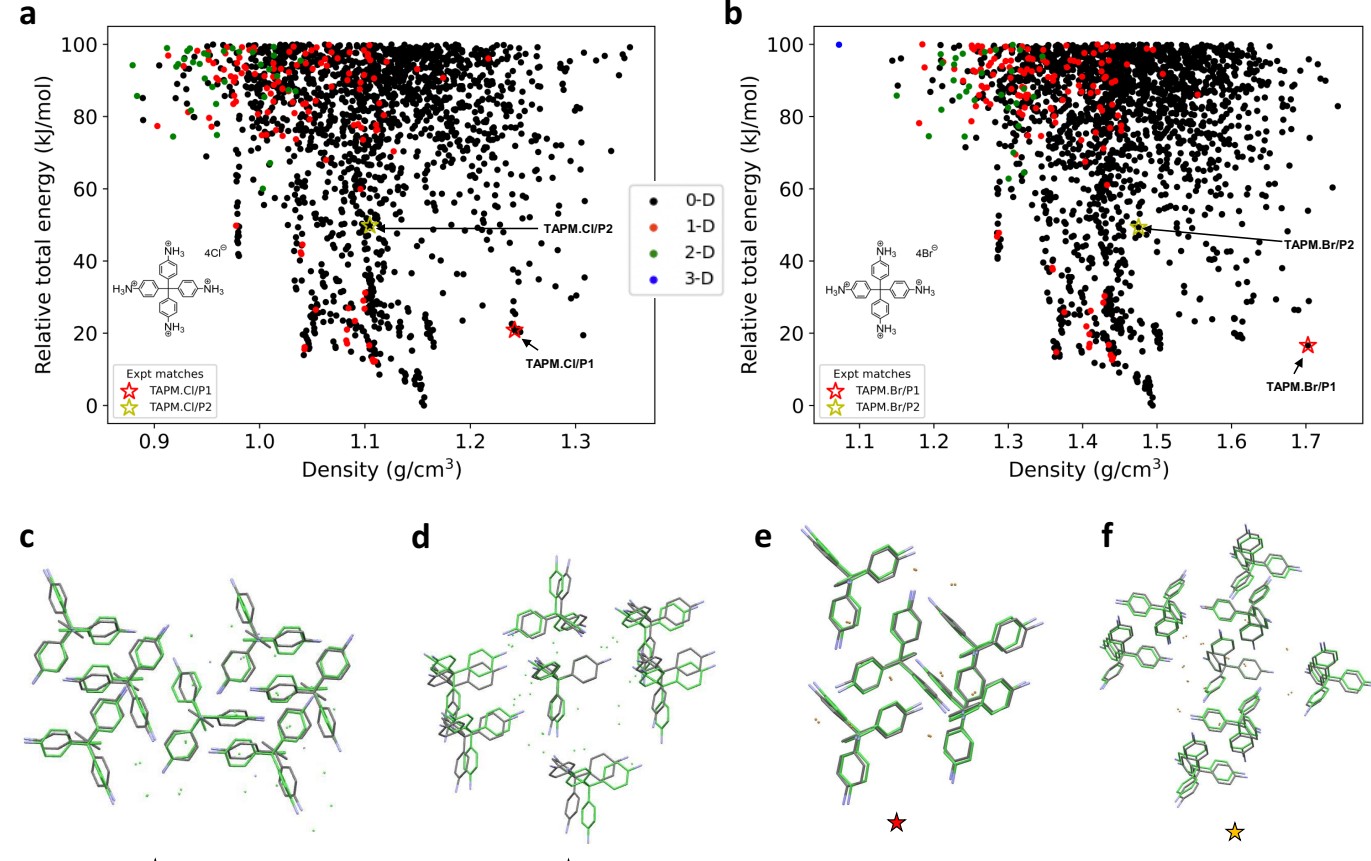

**Extended Data Fig. 1 | Crystal structure prediction suggests dense, non-porous tetrahedral ammonium halide salts.** Energy–density landscapes of the CSP structures for **a**, **TAPM.Cl** and **b**, **TAPM.Br** coloured by dimensionality of pores within each structure. Black data points (*i.e.*, most structures) are either non-porous or contain isolated cavities. Overlay images for experimental structures (coloured by element) and CSP structures (green) for the two experimentally accessible polymorphs of these two salts; **c**, **TAPM.Cl/P1**, **d**, **TAPM.Cl/P2**, **e**, **TAPM.Br/P1** and **f**, **TAPM.Br/P2**. The structural agreement between CSP and experiment is improved by re-optimization of the predicted structures using density functional based tight binding DFTB), which reranks **TAPM.Cl/P1** as the second lowest energy predicted structure (Supplementary Fig. 28). The higher energies of the P2 polymorphs can be explained because these experimental structures are solvates (Supplementary Fig. 2), and solvent is not considered here.

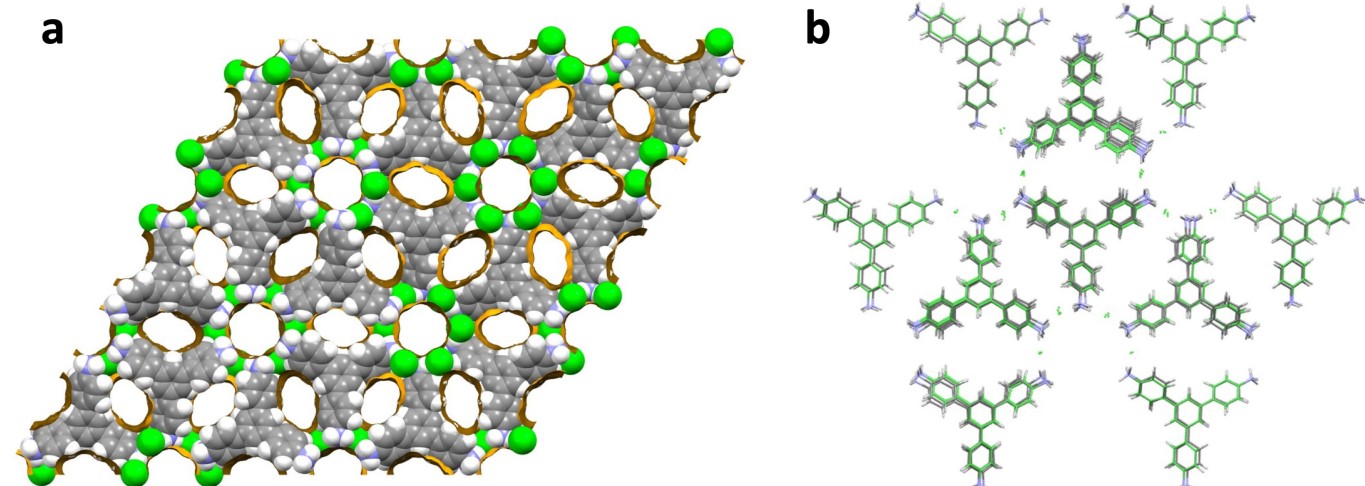

**Extended Data Fig. 2 | A non-metal organic framework by computational design. a**, Structure of **TAPT.Cl**, as predicted using CSP (red star in Fig. 2a). The accessible pore surface is shown in yellow (1.2 Å probe diameter); **b**, Overlay of the experimental single crystal structure (atom colouring by element), obtained for a partial solvate, and the CSP derived structure (green) for **TAPT.Cl**.

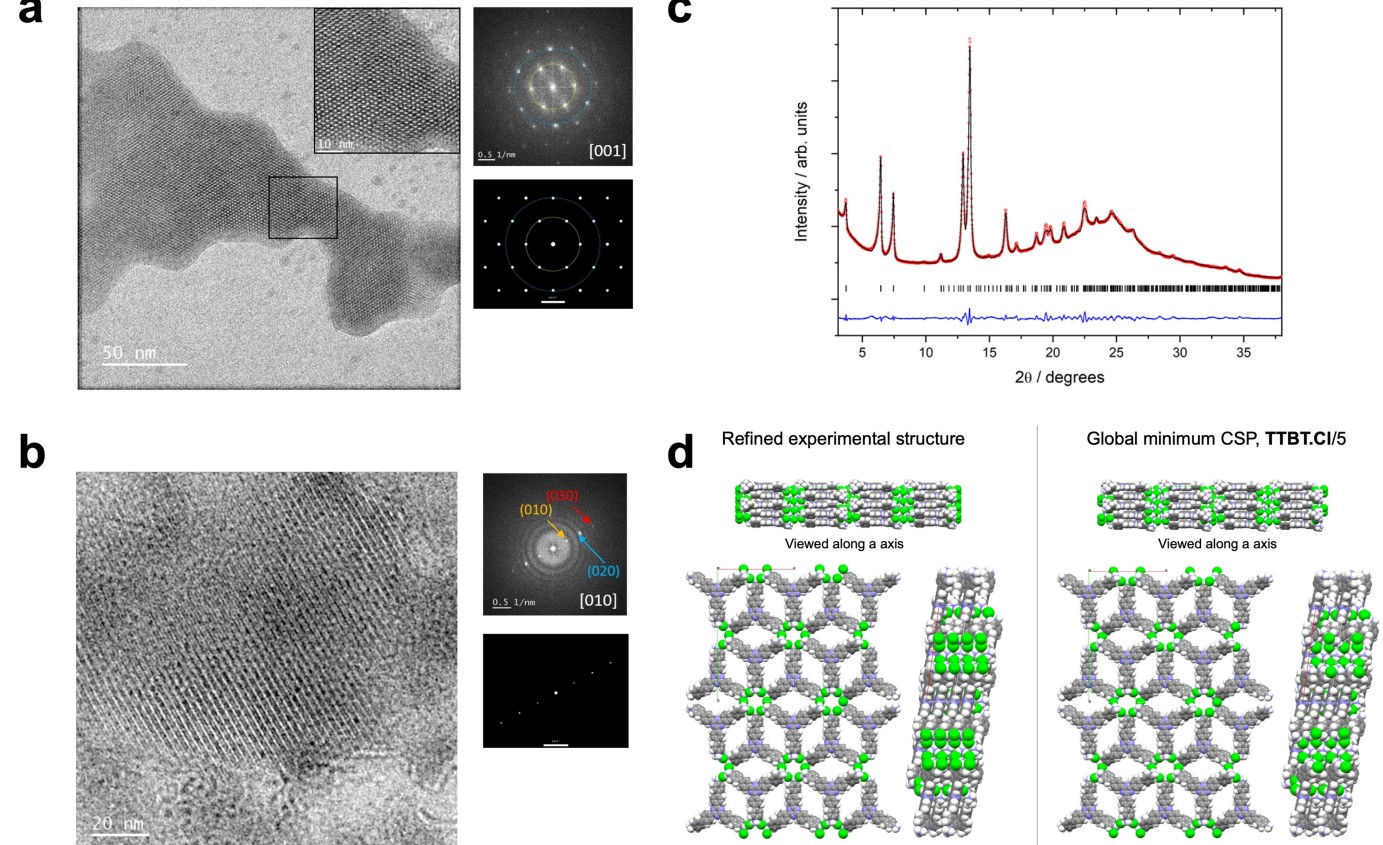

**Extended Data Fig. 3 | Structural characterisation of TT.Br and TTBT.Cl.**
**a**, HR-TEM images of **TT.Br**; the FFT of the TEM data (top right) is shown with the simulated electron diffraction pattern (bottom right) for the best-matched CSP structure (Fig. 3b), aligned in the zone axis [001] (bottom). **c**, HR-TEM images of **TTBT.Cl**; the FFT of the TEM data (top right) is shown, highlighting the (010), (020) and (030) planes, with the simulated electron diffraction pattern (bottom right) for the best-matched CSP structure (Fig. 3c), aligned in

the zone axis [001]. **c**, Observed (red circles), calculated (black line) and difference (blue line) PXRD profiles for structural refinement of a monoclinic model of **TTBT.Cl** ($R_{wp}$ = 1.97 %, $R_p$ = 1.51 %, $\chi^2$ = 3.70). Reflection positions are marked below. Full refinement details are given in the Supplementary Information, Section 2. **d**, Comparison of refined experimental structure for **TTBT.Cl** and global minimum CSP structure, **TTBT.Cl**/5.

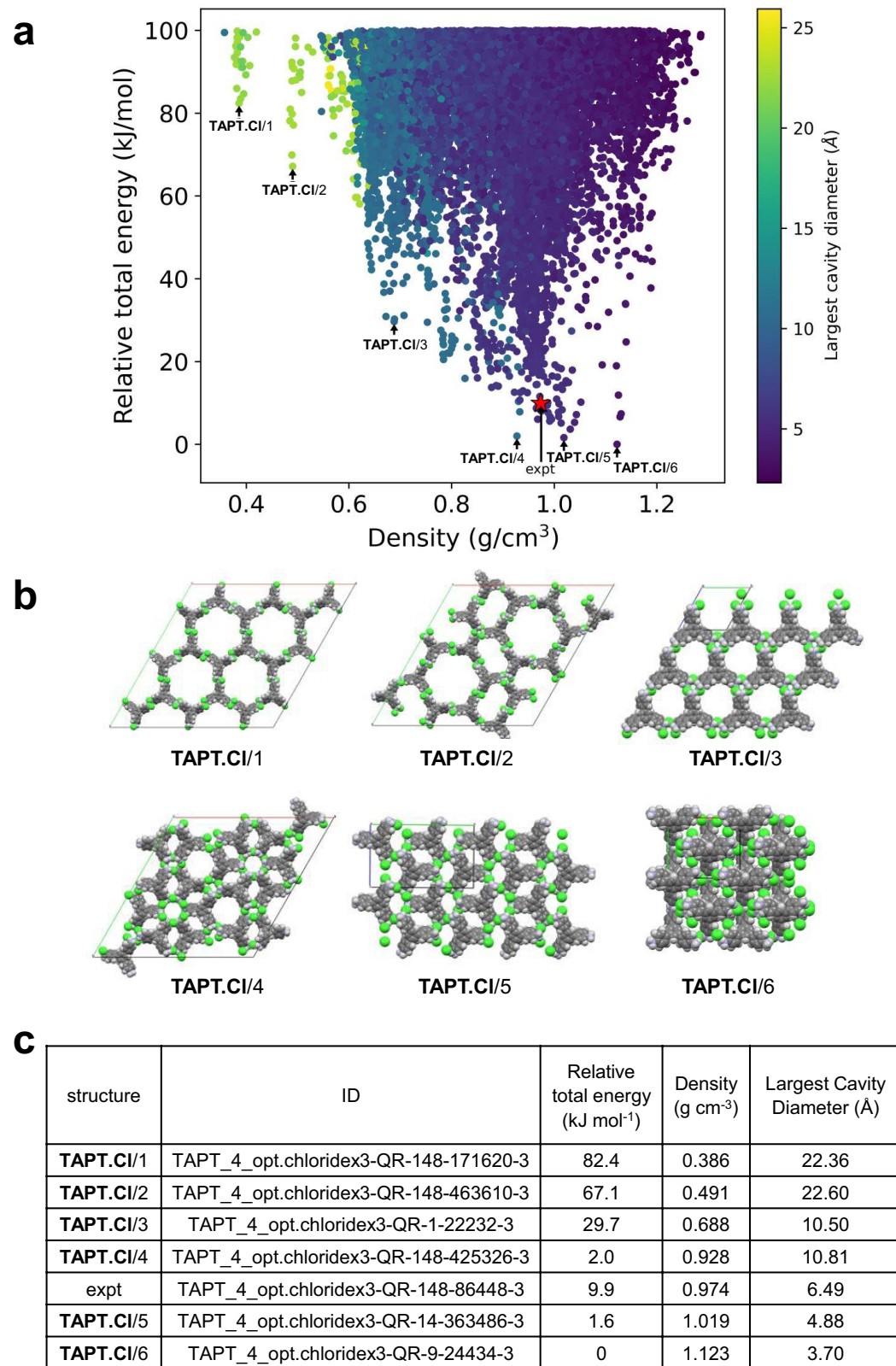

**Extended Data Fig. 4 | Alternative structures on the CSP landscape for TAPT.Cl. a**, Lattice energy plot, as colour-coded by largest cavity diameter, highlighting six alternative packings across a range of physical densities on the leading edge of the landscape for **TAPT.Cl**, focusing on structures in low energy 'spikes'. **b**, Space-filling representations of the six selected **TAPT.Cl** structures. **c**, Table summarizing structure ID's, relative energies relative to the global energy minimum predicted structure, and physical densities of the six highlighted structures, and the observed structure.

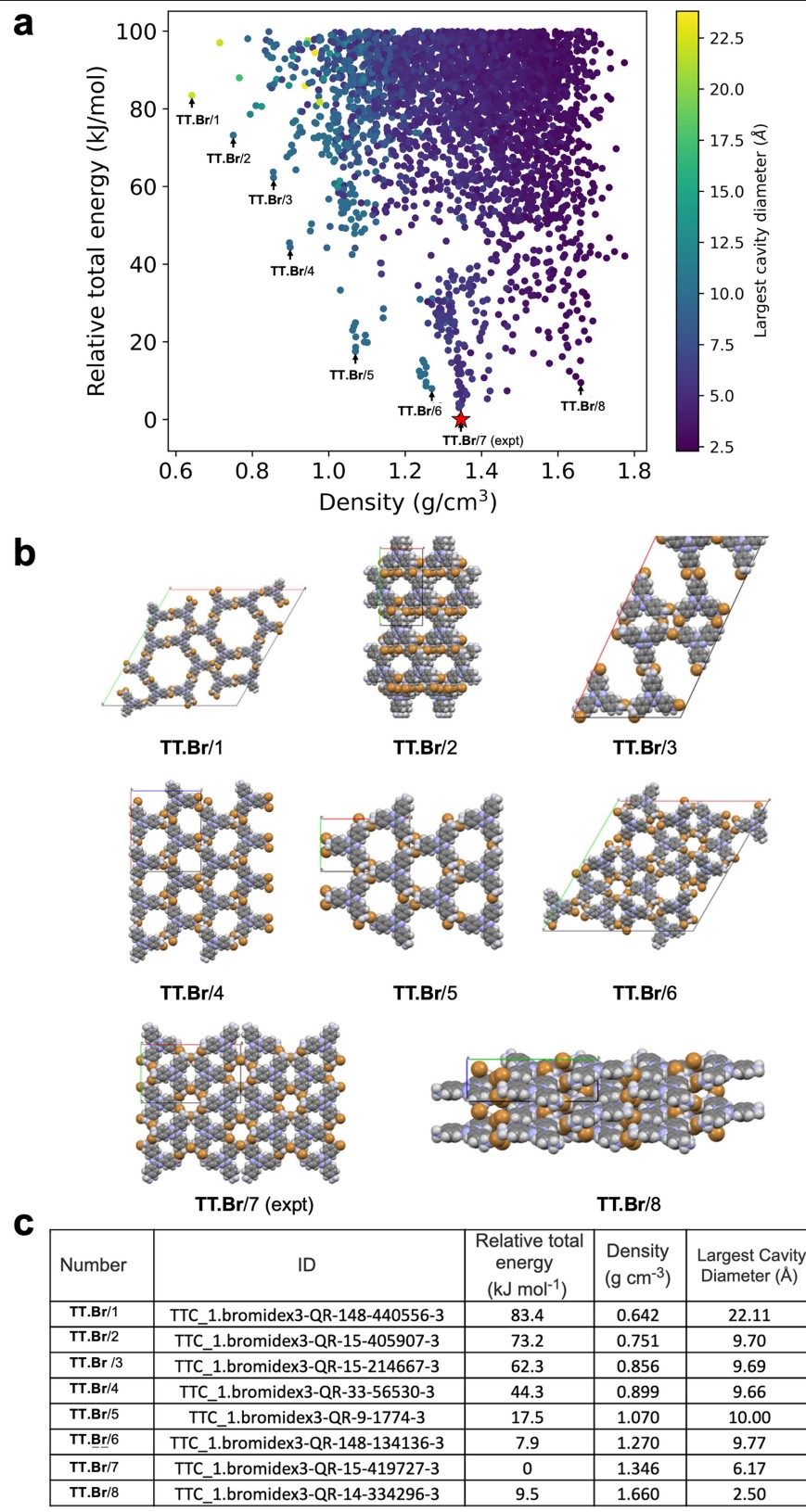

| Number | ID | Relative total energy (kJ mol⁻¹) | Density (g cm⁻³) | Largest Cavity Diameter (Å) |
|---|---|---|---|---|
| **TT.Br**/1 | TTC_1.bromidex3-QR-148-440556-3 | 83.4 | 0.642 | 22.11 |
| **TT.Br**/2 | TTC_1.bromidex3-QR-15-405907-3 | 73.2 | 0.751 | 9.70 |
| **TT.Br** /3 | TTC_1.bromidex3-QR-15-214667-3 | 62.3 | 0.856 | 9.69 |
| **TT.Br**/4 | TTC_1.bromidex3-QR-33-56530-3 | 44.3 | 0.899 | 9.66 |
| **TT.Br**/5 | TTC_1.bromidex3-QR-9-1774-3 | 17.5 | 1.070 | 10.00 |
| **TT.Br**/6 | TTC_1.bromidex3-QR-148-134136-3 | 7.9 | 1.270 | 9.77 |
| **TT.Br**/7 | TTC_1.bromidex3-QR-15-419727-3 | 0 | 1.346 | 6.17 |
| **TT.Br**/8 | TTC_1.bromidex3-QR-14-334296-3 | 9.5 | 1.660 | 2.50 |

**Extended Data Fig. 5 | Alternative structures on the CSP landscape for TT.Br. a**, Lattice energy plot, as colour-coded by largest cavity diameter, highlighting the observed polymorph (**TT.Br**/7) and seven alternative packings across a range of physical densities on the leading edge of the landscape for **TT.Br**, focusing on structures in low energy 'spikes'. **b**, Space-filling representations of the 8 selected **TT.Br** structures. **c**, Table summarizing structure ID's, relative energies relative to the global energy minimum predicted structure, and physical densities of the eight highlighted structures.

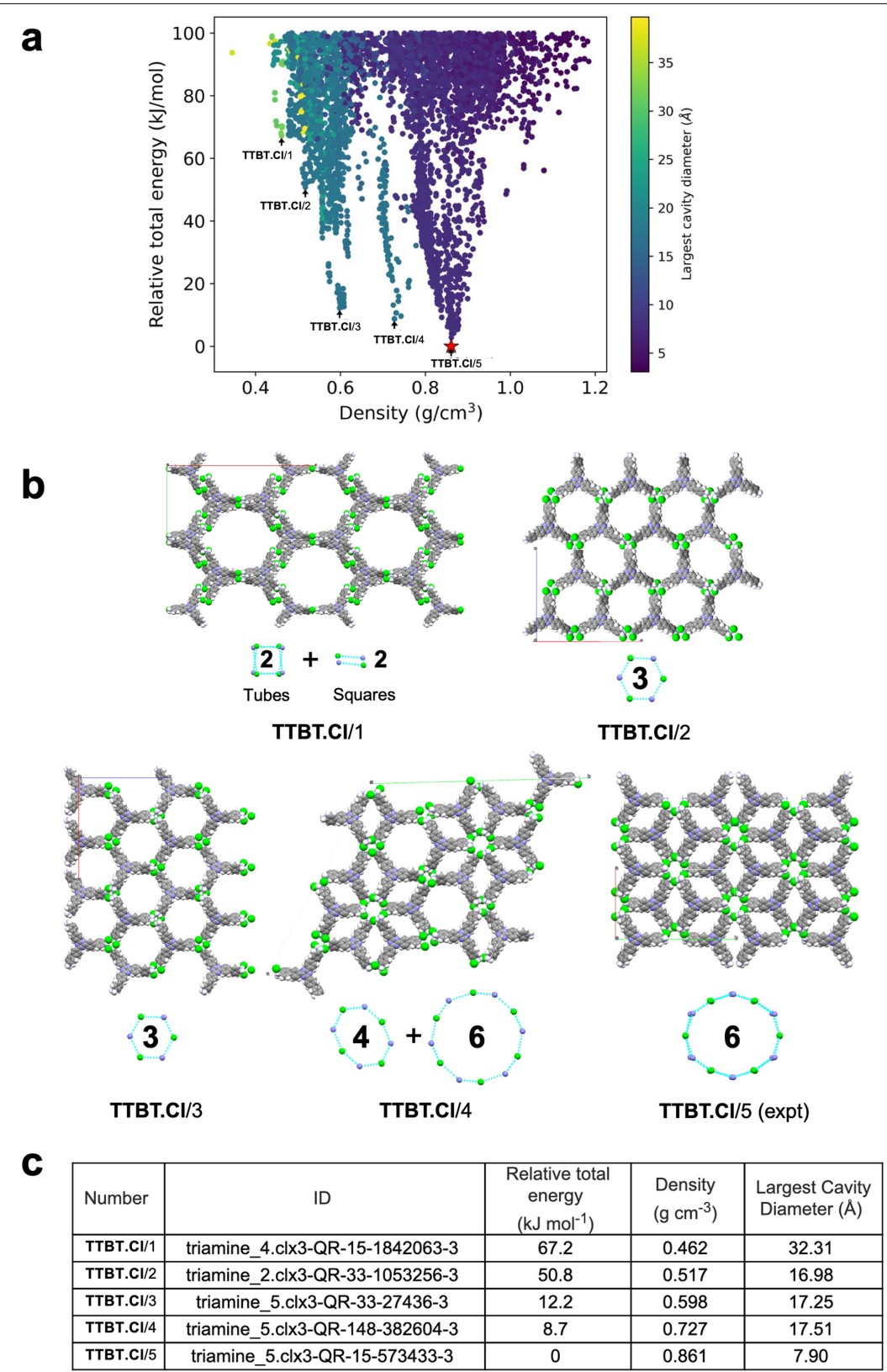

| Number | ID | Relative total energy (kJ mol$^{-1}$) | Density (g cm$^{-3}$) | Largest Cavity Diameter (Å) |
|---|---|---|---|---|
| **TTBT.Cl**/1 | triamine_4.clx3-QR-15-1842063-3 | 67.2 | 0.462 | 32.31 |
| **TTBT.Cl**/2 | triamine_2.clx3-QR-33-1053256-3 | 50.8 | 0.517 | 16.98 |
| **TTBT.Cl**/3 | triamine_5.clx3-QR-33-27436-3 | 12.2 | 0.598 | 17.25 |
| **TTBT.Cl**/4 | triamine_5.clx3-QR-148-382604-3 | 8.7 | 0.727 | 17.51 |
| **TTBT.Cl**/5 | triamine_5.clx3-QR-15-573433-3 | 0 | 0.861 | 7.90 |

**Extended Data Fig. 6 | Alternative structures on the CSP landscape for TTBT.Cl. a**, Lattice energy plot, as colour-coded by largest cavity diameter, highlighting the observed polymorph (**TTBT.Cl**/5) and four alternative packings on the leading edge of the landscape for **TTBT.Cl**, focusing on structures in low energy 'spikes'. **b**, Space-filling representations of the five selected structures. These structures are defined by ionic tubes and rings comprising 6 chlorides (**TTBT.Cl**/5, **TTBT.Cl**/4), 4 chlorides (**TTBT.Cl**/4), 3 chlorides (**TTBT.Cl**/3, **TTBT.Cl**/2) and 2 chlorides (**TTBT.Cl**/1). Equivalent motifs can be found on the structure landscapes for the other two trigonal linkers; for example, **TT.Br**/7 (Extended Data Fig. 5) and has 6-chloride rings, **TTBr**/6 and **TAPT.Cl**/4 (Extended Data Fig. 4) both have a mixture of 4- and 6-chloride rings (c.f., **TTBT.Cl**/4). Also, **TAPT.Cl**/3, **TT.Br**/5 and **TT.Br**/4 are all predicted to form 3-chloride rings (Extended Data Fig. 7). **c**, Table summarizing structure ID's, relative energies, and physical densities of the five highlighted **TTBT.Cl** structures.

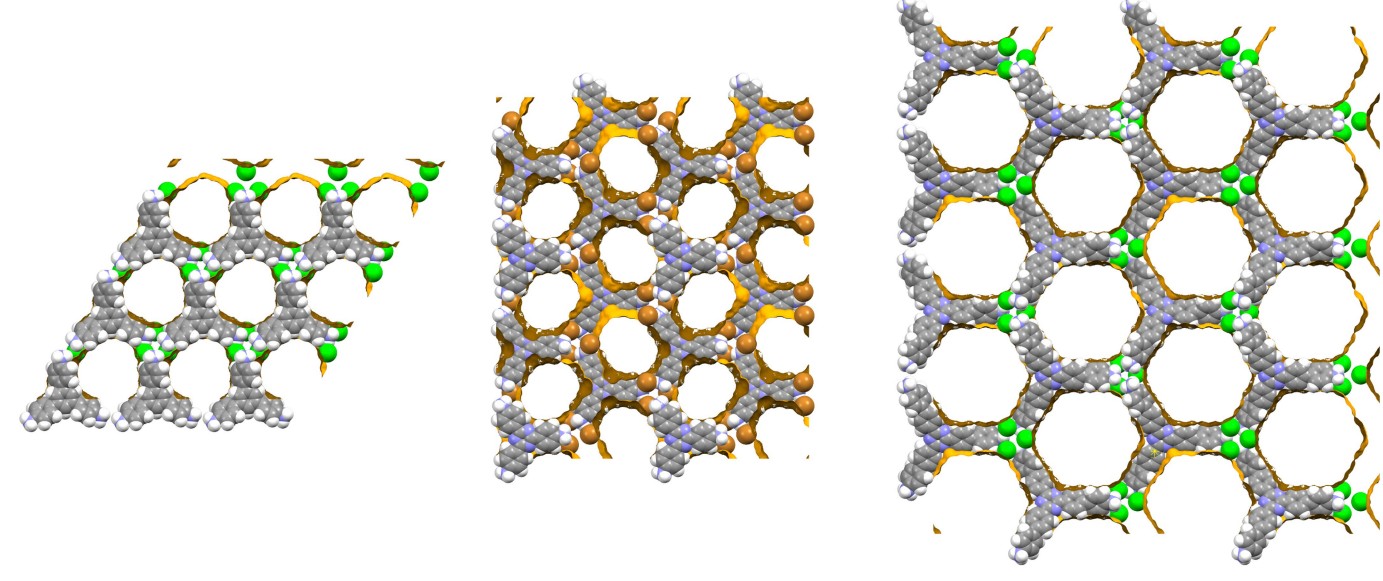

**TAPT.Cl/3**                    **TT.Br/4**                    **TTBT.Cl/3**

**Extended Data Fig. 7 | Hypothetical isoreticular non-metal organic frameworks found on the CSP landscapes for three different organic salts.** These structures were not observed experimentally, despite investigating multiple solvent conditions (Supplementary Information, Section 2), but they all fall within a lattice energy window that suggests that they could in principle be experimentally accessible (<40 kJ mol⁻¹ above global energy minimum, see Extended Data Figs. 4–6). All three structures are defined by ionic tubular pores comprising 3 chlorides, unlike the larger ionic pore (A) observed in the three experimental structures (Fig. 3f), which is formed by 6 chloride anions. The orange pore surface was calculated using a 1.2 Å probe in each case.

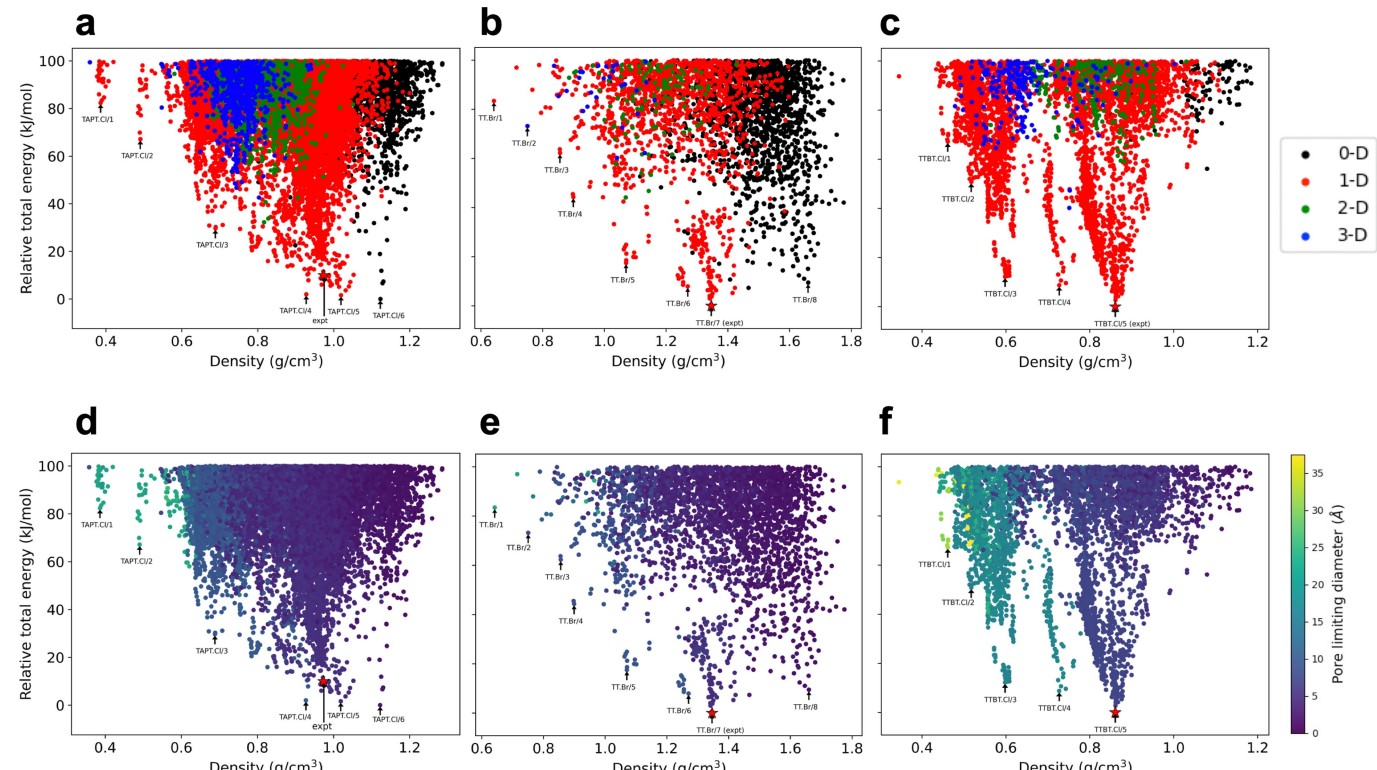

**Extended Data Fig. 8 | Global property analysis using energy-structure-function maps.** The plots show analysis of pore channel geometry and pore limiting diameter for the CSP landscapes of the three trigonal ammonium halide salts. **a**–**c**, CSP landscapes colour coded for pore channel geometry for **a**, **TAPT.Cl**, **b**, **TT.Br** and **c**, **TTBT.Cl**. **d**–**f**, CSP landscapes colour coded for pore limiting diameter for **a**, **TAPT.Cl**, **b**, **TT.Br** and **c**, **TTBT.Cl**. All channel dimensionality calculations used a probe radius of 1.65 Å.

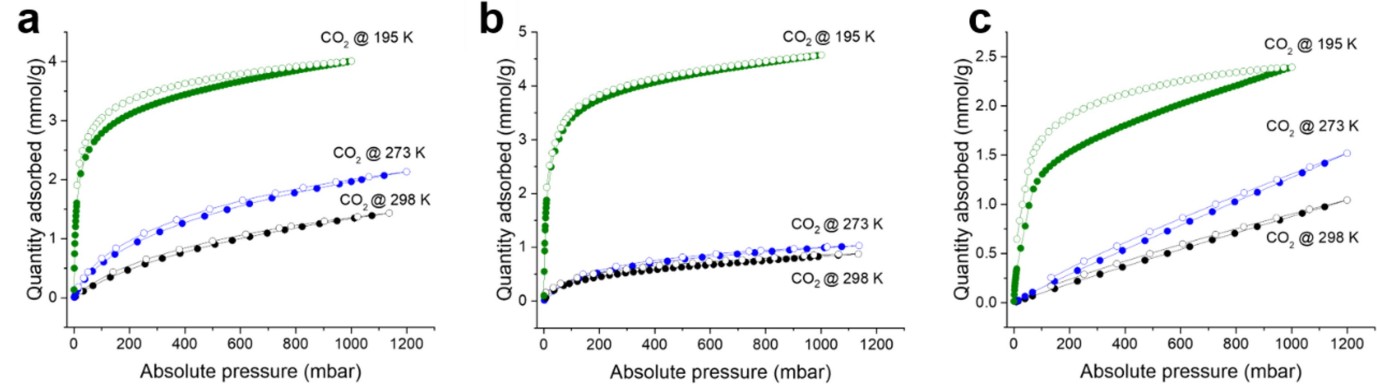

**Extended Data Fig. 9 | Carbon dioxide isotherms for porous salts at different temperatures. a**, **TAPT.Cl**, **b**, **TT.Br** and **c**, **TTBT.Cl**. Filled symbols are for adsorption isotherms, open symbols are for desorption isotherms.