## [Peer Review File · Nature]

Manuscript Title: Porous isorecticular non-metal organic frameworks

Reviewer Comments & Author Rebuttals

Reviewer Reports on the Initial Version:

Referees' comments:

Referee #1 (Remarks to the Author):

Review of manuscript ms-2023-11-20963

O'Shaughnessy and co-workers report the development of a new design strategy for porous isorecticular and metal-free framework materials. Specifically, the authors use state-of-the-art crystal structure prediction methods to identify rigid polyamine molecules whose chloride and bromide salts form *permanently porous* and *crystalline* isorecticular framework structures, which is an important and unique hallmark of metal-organic frameworks (MOFs).¹ It has been also shown that these organic salts have remarkable radioiodine capture abilities, which exceed those of most MOFs. The reported findings are highly relevant to the fields of materials science, supramolecular chemistry, computational chemistry, organic chemistry, crystal engineering, environmental chemistry and to solid-state sciences in general. The manuscript has all the qualities of a landmark paper.

The study is very comprehensive, covering the synthesis of the target compounds, the extensive characterisation of the prepared solids, computational crystal structure predictions (and other computational analyses), as well as gas sorption and iodine uptake studies. It is obvious that the experiments were thoughtfully and carefully planned; the data appears to be meticulously and expertly analysed. The authors have also very thoroughly assessed the state of the art, and support their claims objectively through the citation of relevant studies. There are several factors that make the study stand out, and render it extremely important and of interest to a broad audience of scientists:

1) At present, there is enormous interest in porous materials due to their potential applicability in a number of fields including catalysis, separation sciences and the storage of energy and gases. Besides the heavily commercialised zeolites, MOFs (and more recently covalent organic frameworks, COFs) are currently among the most studied class of porous materials and are generally regarded as most promising for a range of industrial applications. Although both MOFs and COFs are highly functional,² amenable to reticular chemistry,¹ scalable and processable,³ their wider use is currently constrained by a range of material-specific characteristics. In the case of MOFs, it is the generally inadequate stability^{4,5} and the frequent use either toxic or expensive metals, while COFs suffer from poor(er) crystallinity, stability and limited structural variety.⁶ This naturally prompts the need to further diversify the landscape of porous materials that are easily accessible, stable, affordable and crystalline. And organic salts emerge as a particularly appealing choice: they have been previously studied in this context and have been shown to be good supramolecular host-guest systems.^{7,8} But

they have not been developed at the same pace as the ever-popular MOFs. Also, no class of organic salts has, so far, been identified as a likely starting point for the formation of permanently porous crystals.

2) The design and development of crystalline and permanently porous isorecticular organic salts is a remarkably difficult task, and it is more remarkable still that the authors achieved this using ammonium halides – it is not obvious that polyamines could form such materials in simple reactions with ubiquitous inorganic acids. The challenges affiliated with the use ammonium halides stem from our poor ability to design molecular solid-state structures using non-directional ionic interactions.⁹ The use of ammonium halides to prepare the targeted solids would therefore *usually* have required extensive (and often ineffective) screening experiments. To avoid this, the authors took advantage of crystal structure prediction (CSP) methods to guide the development of the target compounds. The successful use of CSP in the engineering of porous isorecticular organic salts is also a significant achievement in itself, as the last two Crystal Structure Prediction Blind Tests have shown that organic salts are exceedingly challenging to predict.^{10,11} The authors' computational studies have not only permitted the identification of promising molecular tectons, but also allowed them to correlate the stability of putative crystal structures with the size and the topology of the molecular tecton. The use of CSP in developing porous isorecticular molecular salts in this manner heightens the uniqueness of this study, and is a very exciting advancement in our understanding of molecular solid-state structures, and our ability to design them.

3) The presented results are likely to trigger great interest from various scientific communities, as they suggest that porous organic salts can compete with MOFs in terms of functionality. The ease with which these crystals were synthesised might also invite many other research groups to investigate this class of solids. I strongly believe that these materials will not only capture the attention of materials scientists, solid-state chemists and crystal engineers, but also that of organic synthetic chemists and process chemists (scale-up and commercialisation), separation scientists, environmental chemists (gas sorption), chemical biologists (biocatalysis and separation), pharmaceutical formulation scientists (drug delivery) and possibly, the community of researchers interested in water harvesting (ammonium halides tend to be very hygroscopic).

This proof-of-principle study is well rounded, and I don't see any need to add much more to it. I do have, however, some comments and suggestions that the authors might want to consider:

a) The authors highlighted how well ammonium halides are known in the context of pharmaceuticals, and this begs the question whether the reported solids are as hygroscopic as pharmaceutical halides are known to be. Dynamic vapour sorption (DVS) studies should be able to provide more insight into this the stability of these porous ammonium halides. I note that the results of such DVS studies should **not** be a critical factor in deciding whether the manuscript should be published in Nature. A high stability towards hydration would make the prepared solids even more attractive than they already are, while a lower stability would indicate the extent to which these materials would need to be protected from atmospheric water in a marketed device.

b) The stacks of diffractograms shown in Figures 2g-i don't contain the difference curves, which could leave the reader under the impression that the PXRD structural analyses are incomplete. It is

very obvious that this is not the case, as one immediately realises after reading the corresponding (very thorough) section in the Supplementary Information (SI) document. I suggest the authors make it clear (either in the figure caption or the main text) that the analysis was not straightforward, and that a full account of the PXRD analysis is given in the SI document. It might also be helpful to add a comment to this section about the feasibility of including the difference curves to the stacks of diffractograms.

c) The main text suggests that the tetrahedral tectons are not promising candidates for the formation of porous salts. I note, however, that the energy-density landscapes shown in Extended Data Figure 1a reveals a spike of lower energy structures, some of which feature channels. Could such structures be accessed using various bulky template molecules,¹² which could subsequently be removed? Would this activated solid be stable, or would it collapse? Is it worthwhile pursuing such structures? I do not expect that this is addressed experimentally on this occasion, but it might be something that should be considered when discussing such landscapes.

d) Page 1, Line 16: The 'metal-organic bonding' is better described as 'coordinate covalent bonding'.

e) Page 2, Lines 1-8: I would describe hydrogen-bonded organic frameworks (HOFs) as a type of extended framework structure. I agree that HOFs are held together by weaker interactions than those seen in MOFs and COFs, but despite this, HOFs are, in my opinion, still an example of an extended framework material.

f) Page 7, Line 13: I suggest changing '14.3 × 8.5 Å' into '14.3 Å × 8.5 Å' or '14.3 × 8.5 Å²'

g) Page 7, L32: How was the isostructurality quantified?^{13,14}

h) Page 8, Line 1: A part of the sentence seems to be missing: 'as used to provide structural'.

i) SI, Pages 7-8: The term 'good' solvent is well established in the community of solid-state chemists, but a broader readership might be better served with 'effective solvent'.

j) SI, Page 25, Figure 18: The authors suggest that the changes in the PXRD of TTBT.Cl before and after iodine exposure are not significant. I note that the patterns are similar only up to about 7° 2θ. Differences in the 7-50° range are rather significant.

k) SI document, Section 3: The alpha letter is capitalised. Please change to small caps. Also, various quantities and parameters (e.g., *F* and *R*₁) should be italicised. Some numbers and symbols should be superscripted (e.g., mol⁻¹)

l) Dash symbols are used instead of the mathematical minus symbol throughout the text and in figures. Please correct this.

m) The units are inconsistently presented: kJ/mol and g cm⁻³. I suggest the use of kJ mol⁻¹ and g cm⁻³.

In conclusion, the study is a major milestone and truly innovative, and it is also exemplary in its execution. The authors have made an exceptional advance in the area of porous materials and materials science, and I look forward to seeing how the scientific community responds to their work.

Dr Dejan-Krešimir Bučar
Department of Chemistry
University College London
London, UK

References

1. O. M. Yaghi, M. O'Keeffe, N. W. Ockwig, H. K. Chae, M. Eddaoudi, J. Kim, Reticular synthesis and the design of new materials. *Nature* 2003, 423, 705.
2. R. Freund, O. Zaremba, G. Arnauts, R. Ameloot, G. Skorupskii, M. Dincă, A. Bavykina, J. Gascon, A. Ejsmont, J. Goscińska, M. Kalmutzki, U. Lächelt, E. Ploetz, C. S. Diercks, S. Wuttke, The current status of MOF and COF applications. *Angew. Chem. Int. Ed.* 2021, 60, 23975.
3. D. Crawford, J. Casaban, R. Haydon, N. Giri, T. McNally, S. L. James, Synthesis by extrusion: continuous, large-scale preparation of MOFs using little or no solvent. *Chem. Sci.*, 2015, 6, 1645.
4. P. Guo, D. Dutta, A. G. Wong-Foy, D. W. Gidley and A. J. Matzger, Water sensitivity in Zn₄O-based MOFs is structure and history dependent, *J. Am. Chem. Soc.* 2015, 137, 2651.
5. C. Mottillo and T. Friščić, Carbon dioxide sensitivity of zeolitic imidazolate frameworks, *Angew. Chem. Int. Ed.*, 2014, 53, 7471.
6. F. Haase and B. V. Lotsch, Solving the COF trilemma: Towards crystalline, stable and functional covalent organic frameworks, *Chem. Soc. Rev.* 2020, 49, 8469.
7. N. Malek, T. Maris, M. Simard, J. D. Wuest, Molecular tectonics. Selective exchange of cations in porous anionic hydrogen-bonded networks built from derivatives of tetraphenylborate. *J. Am. Chem. Soc.* 2005, 127, 5910.
8. V. A. Russell, M. C. Etter, M. D. Ward, Layered materials by molecular design: structural enforcement by hydrogen bonding in guanidinium alkane- and arenesulfonates, *J. Am. Chem. Soc.*, 1994, 116, 1941.
9. M. K. Corpinot, D.-K. Bučar, A practical guide to the design of molecular crystals. *Cryst. Growth Des.* 2019, 19, 1426.
10. D. A. Bardwell et al. Towards crystal structure prediction of complex organic compounds - a report on the fifth blind test. *Acta Cryst. Sect. B*, 2011, 67, 535.
11. M. A. Reilly et al. Report on the sixth blind test of organic crystal structure prediction methods. *Acta Crystallogr. Sect. B*, 2016, 72, 439.
12. K. Travis Holman, Stephen M. Martin, Daniel P. Parker, and Michael D. Ward The generality of architectural isomerism in designer inclusion frameworks, *J. Am. Chem. Soc.* 2001, 123, 442.
13. A. Kálmán, L. Párkányi, G. Argay, Classification of the isostructurality of organic molecules in the crystalline state. *Acta Cryst. Sect. B*, 1993, 49, 1039.
14. P. Bombicz, What is isostructurality? Questions on the definition. *IUCrJ*, 2023, 11, 3.

Referee #2 (Remarks to the Author):

In this manuscript the Authors presented the first time application of crystal structure prediction (CSP) to crystalline porous organic salts (CPOS). In these materials salt anions take the role of nodes, connected by positively charged organic linkers, a charge reversal compared to the nodes and linkers found in MOFs.

The key novelty of this work is the demonstration that CSP can reliably predict the structures of CPOS materials. The success of the method was tested on four linker types in combination with two halide anions, resulting in eight predicted energy landscapes. In all cases, the experimentally-observed structures were matched against the low energy predicted structures. Clearly CSP is a powerful and robust method for designing such materials computationally. Nonetheless, I would like to point at several aspects of the study, which, in my opinion need to be improved:

1) The key difference between MOFs and CPOS is not only the reversal of charges on nodes and linkers, but a different mode of binding between nodes and linkers: covalent coordination bonds in MOFs and non-covalent ionic interactions in CPOS. For one thing, this leads to fundamental differences in the CSP approaches needed for both of these types of materials, which would be useful to discuss in the manuscript.

From a material stability point of view, it would be interesting to quantify and compare the energies of these ionic interactions with the typical coordination bonds found in MOFs. How does that reflect on the thermal stability of CPOS, as well as mechanical properties of these materials?

2) The synthesized materials demonstrate favorable iodine sorption performance. However, it is recognized in the manuscript that calculation of pore volume in the predicted structures is not a quantitatively-accurate predictor for sorption capacity. Perhaps some GC-MC/MD simulations, or some alternative method, could be applied, at least for selected structures, to show whether the iodine sorption capacity can be predicted quantitatively.

3) I would like to ask to replace PXRD comparison figures, where the pattern simulated from the predicted structure is shown above the experimental pattern, with Rietveld refinement plots. Showing the agreement between the experimental and calculated (refined) profile, together with the difference curve would be a stronger proof for the agreement between predicted and synthesized structures.

This is particularly necessary for the structure of TTBT.Cl, where simulated and experimental peak intensities are widely different. Rietveld refinement could clarify, whether this is caused by preferred orientation, presence of guest molecules in the pores or issues with the geometry of the structure itself.

The description of the PXRD analysis of TTBT.Cl got me somewhat confused. The Authors report indexing the powder pattern with a monoclinic C-centered cell: $a=27.581$, $b=47.314$, $c=8.0051$, $\beta = 97.61$, saying that this cell is similar to that of the lowest energy predicted structure. But looking at the CIF of the predicted structure I found a cell with dimensions:

$a=80.60$, $b=40.11$, $c=6.16$, $\beta=90.06$, which is very different. This needs to be clarified.

4) Related to the previous point, TTBT.Cl is clearly the most challenging system of all presented. The

linker is conformationally flexible, there is a possibility of the experimental structure having a different space group symmetry from the predicted model. I think some of these ambiguities could be resolved by a more detailed analysis of the experimental PXRD pattern, starting from the model of the predicted structure, rather than attempting full ab initio structure solution. Whilst refinement of all atom positions is not feasible, given the low data quality, perhaps molecular fragments defined via z-matrix with some flexible torsion angles could be used?

To summarize, I enjoyed reading the manuscript and I believe that this work represents a major milestone for the design of porous materials. The article is written well, the methodology and the results are clearly explained.

Referee #3 (Remarks to the Author):

Review of "Porous isorecticular non-metal organic frameworks"
O'Shaughnessy, et al.

This manuscript by O'Shaughnessy, et al. describes a combined computational and experimental approach to the synthesis of porous organic salts, analogous to MOFs but flipping the script with respect to the charge on the node (negatively charged halide ions) and linkers (positively charged polyvalent ammonium ions). I've had the opportunity to review many MOF articles in which computations, either data mining or ab initio methods, are used to predict new structures and sometimes their adsorption properties. The O'Shaughnessy submission, however, is a rare example of computations confirmed by experiment. Here, CSP is used to identify the lowest energy polymorphs and then the authors actually crystallized the lowest energy forms for two of the linkers and one of the lowest for a third (i.e., near the low energy" tip of the CSP grouping. This is unique, and in my view will be of interest to the broader community, particularly those interested in de novo design of organic solids as well as those interested in using CSP reliably to identify polymorphs. The authors posit sound reasoning that explains the relationship between packing density and the coordination number of the linker groups about the spherical anionic nodes. I would have been supportive of publication if the manuscript ended here, but the authors also demonstrated iodine adsorption, which is reversible with respect to regenerating the original host structure. This is a phenomena that seems deserving of a more detailed investigation, but it is sufficient for this submission as it demonstrates clearly that using CSP to accelerate synthesis of a functional material, here in a relatively unexplored class of materials, can have a useful application (here new materials for radioactive iodine capture). I strongly support publication. The conclusions are well-supported by the data, and the manuscript is well-written with clarity and a logical flow. I have some questions/comments, however, offered here in the spirit of improving clarity.

(1) A comment about the term "HOF". There is a lack of clarity and a lot of confusion about this term, because it was first coined by Banglin Chen to include hydrogen-bonded organic frameworks that were porous. This led to labeling them as HOF-1, HOF-2, etc., I believe in Y2013. But of course hydrogen-bonded organic frameworks were known decades before. I like that the authors have largely used the POS alternative, although this would not apply to neutral compounds.

(2) Regarding point (1), in my view, claims of porosity in HOFs is tenuous in many examples. The authors correctly note that many HOFs are not stable. One example that is irrefutable is Brekalo, et al. (Angew. Chem. Int. Ed. 2020, 59, 1997–2002, DOI: 10.1002/anie.201911861), which describe guanidinium benzenedisulfonate, which is truly porous and was distinguished from HOFs in general using the term p-HOF. The authors may want to cite Brekalo, et al. in the introduction among their example POS materials.

(3) At the risk of revealing my identity (in any case I signed my review below), I would like to note that polymorphism has never been observed among the more than 700 crystalline guanidinium organosulfonate compounds (either guest-filled or guest free). These compounds are salts. Some of these compounds have polyvalent sulfonate “linkers” connected to guanidinium ions through hydrogen bonding, some with crystal structures have topologies similar to those described by O’Shaughnessy, although densely packed because of guest occupation or interdigitation of organosulfonate residues. But can the absence of polymorphism in guanidinium organosulfonates be explained by the directional nature of the hydrogen bonds between guanidinium ions and sulfonate groups? Conversely, would negatively charged sulfonates linked to spherical cations be more likely to exhibit polymorphism because of the absence of directional bonding? I have never considered the link between directional hydrogen bonding and the absence of polymorphism, and I would be very interested to know if the authors have considered it. If so, maybe worth mentioning in the article?

(4) Page 7, line 6: “Figure 2a-c. In all three cases, these matches were found to lie at the tip of a ‘spike’ in the CSP landscape, and for TT.Br and TTBT.Cl these structures corresponded to the predicted global “ energy minimum structures. Again, this is a remarkable correspondence of CSP and experimental outcome.

(5) Page 7, line 9: Maybe add some clarity to this statement: “These three crystal packings were all isostructural and comprised two distinct one-dimensional (1-D) pore channels, as labelled in Figure 2d-f.” At this stage of the manuscript, crystal packings in the experimental structures were deduced from CSP and comparison to PXRD, not single crystal X-ray diffraction (later, single crystal X-ray diffraction confirmed one of the structures).

(6) The “partial solvate” in TAPT.Cl is a bit indefinite. Can the authors state the amount of solvent, and the true porosity of the single crystal when solvent is accounted for?

(7) Regarding iodine adsorption, it may be a good idea to express the amount of adsorption as mole iodine/mol host, as mol/g may not be so important for the application stated here, and mol/mol is more helpful to understanding the occupancy of iodine in the host.

Michael D. Ward

Author Rebuttals to Initial Comments:

Referees' comments:

We thank all three referees for their thoughtful and constructive comments.
We respond in detail to each specific point below.

Referee #1 (Remarks to the Author):

Review of manuscript ms-2023-11-20963

O'Shaughnessy and co-workers report the development of a new design strategy for porous isorecticular and metal-free framework materials. Specifically, the authors use state-of-the-art crystal structure prediction methods to identify rigid polyamine molecules whose chloride and bromide salts form *permanently porous* and *crystalline* isorecticular framework structures, which is an important and unique hallmark of metal-organic frameworks (MOFs).¹ It has been also shown that these organic salts have remarkable radioiodine capture abilities, which exceed those of most MOFs. The reported findings are highly relevant to the fields of materials science, supramolecular chemistry, computational chemistry, organic chemistry, crystal engineering, environmental chemistry and to solid-state sciences in general. The manuscript has all the qualities of a landmark paper.

The study is very comprehensive, covering the synthesis of the target compounds, the extensive characterisation of the prepared solids, computational crystal structure predictions (and other computational analyses), as well as gas sorption and iodine uptake studies. It is obvious that the experiments were thoughtfully and carefully planned; the data appears to be meticulously and expertly analysed. The authors have also very thoroughly assessed the state of the art, and support their claims objectively through the citation of relevant studies. There are several factors that make the study stand out, and render it extremely important and of interest to a broad audience of scientists:

1) At present, there is enormous interest in porous materials due to their potential applicability in a number of fields including catalysis, separation sciences and the storage of energy and gases. Besides the heavily commercialised zeolites, MOFs (and more recently covalent organic frameworks, COFs) are currently among the most studied class of porous materials and are generally regarded as most promising for a range of industrial applications. Although both MOFs and COFs are highly functional,² amenable to reticular chemistry,¹ scalable and processable,³ their wider use is currently constrained by a range of material-specific characteristics. In the case of MOFs, it is the generally inadequate stability^{4,5} and the frequent use either toxic or expensive metals, while COFs suffer from poor(er) crystallinity, stability and limited structural variety.⁶ This naturally prompts the need to further diversify the landscape of porous materials that are easily accessible, stable, affordable and crystalline. And organic salts emerge as a particularly appealing choice: they have been previously studied in this context and have been shown to be good supramolecular host-guest systems.^{7,8} But they have not been developed at the same pace as the ever-popular MOFs. Also, no class of organic salts has, so far, been identified as a likely starting

point for the formation of permanently porous crystals.

2) The design and development of crystalline and permanently porous isorecticular organic salts is a remarkably difficult task, and it is more remarkable still that the authors achieved this using ammonium halides – it is not obvious that polyamines could form such materials in simple reactions with ubiquitous inorganic acids. The challenges affiliated with the use ammonium halides stem from our poor ability to design molecular solid-state structures using non-directional ionic interactions.⁹ The use of ammonium halides to prepare the targeted solids would therefore *usually* have required extensive (and often ineffective) screening experiments. To avoid this, the authors took advantage of crystal structure prediction (CSP) methods to guide the development of the target compounds. The successful use of CSP in the engineering of porous isorecticular organic salts is also a significant achievement in itself, as the last two Crystal Structure Prediction Blind Tests have shown that organic salts are exceedingly challenging to predict.^{10,11} The authors' computational studies have not only permitted the identification of promising molecular tectons, but also allowed them to correlate the stability of putative crystal structures with the size and the topology of the molecular tecton. The use of CSP in developing porous isorecticular molecular salts in this manner heightens the uniqueness of this study, and is a very exciting advancement in our understanding of molecular solid-state structures, and our ability to design them.

3) The presented results are likely to trigger great interest from various scientific communities, as they suggest that porous organic salts can compete with MOFs in terms of functionality. The ease with which these crystals were synthesised might also invite many other research groups to investigate this class of solids. I strongly believe that these materials will not only capture the attention of materials scientists, solid-state chemists and crystal engineers, but also that of organic synthetic chemists and process chemists (scale-up and commercialisation), separation scientists, environmental chemists (gas sorption), chemical biologists (biocatalysis and separation), pharmaceutical formulation scientists (drug delivery) and possibly, the community of researchers interested in water harvesting (ammonium halides tend to be very hygroscopic).

Response: We thank the Reviewer for this detailed summary and positive comments.

This proof-of-principle study is well rounded, and I don't see any need to add much more to it. I do have, however, some comments and suggestions that the authors might want to consider:

a) The authors highlighted how well ammonium halides are known in the context of pharmaceuticals, and this begs the question whether the reported solids are as hygroscopic as pharmaceutical halides are known to be. Dynamic vapour sorption (DVS) studies should be able to provide more insight into this the stability of these porous ammonium halides. I note that the results of such DVS studies should **not** be a critical factor in deciding whether the manuscript should be published in Nature. A high stability towards hydration would make the prepared solids even more attractive than they already are, while a lower stability would indicate the extent to which these

materials would need to be protected from atmospheric water in a marketed device.

Response: This is a good question that has practical importance. The water stability of these porous ammonium halide salts has now been tested. We have added the following short section to answer the referee's question (pg. 12, line 28 *et seq.*).

“Water stability

The stability of frameworks to water is another important practical consideration. For these ammonium halide salts, this depends on the organic linker. **TAPT.Cl** is water soluble while **TT.Br** has very low water solubility, but becomes amorphous upon immersion in water. By contrast, the more hydrophobic **TTBT.Cl** framework is insoluble in water and a sample submerged in water was shown by PXRD to have stable crystallinity for at least 48 h (Supplementary Fig. S23). Water adsorption isotherms were also collected for **TTBT.Cl** and it was shown to adsorb 12.4 mmol g⁻¹ of water (Supplementary Fig. S24). PXRD analysis before and after water sorption showed that the sample had retained a good level of crystallinity.”

Figure S23. PXRD patterns for a porous **TTBT.Cl** sample that was submerged in water at room temperature, as sampled at different time intervals. By contrast, the porous **TAPT.Cl** framework is soluble in water, and **TT.Br** becomes amorphous upon immersion in water.

Figure S24. Water adsorption isotherm for **TTBT.Cl**. The maximum water uptake is 12.4 mmol g⁻¹.

b) The stacks of diffractograms shown in Figures 2g-i don't contain the difference curves, which could leave the reader under the impression that the PXRD structural analyses are incomplete. It is very obvious that this is not the case, as one immediately realises after reading the corresponding (very thorough) section in the Supplementary Information (SI) document. I suggest the authors make it clear (either in the figure caption or the main text) that the analysis was not straightforward, and that a full account of the PXRD analysis is given in the SI document. It might also be helpful to add a comment to this section about the feasibility of including the difference curves to the stacks of diffractograms.

Response: Thanks for this suggestion. To address this, we have now added a reference to the figure caption as follows (pg. 5, line 15):

“Details of the analyses for **TAPT.Cl** and **TT.Br**, and the refinement for **TTBT.Cl** are given in the Supplementary Information (Section 2).”

Since the CSP calculations are conducted without experimental input, there is unavoidably some discrepancy in the lattice parameters of the predicted structures and those of the realised bulk material. For one thing, CSP does not capture any uncorrelated disorder in structures, which is suggested for **TTBT.Cl** by the halos observed in the FFT TEM data (Extended Data Fig. 3b).

Difference curves based on the simulated profiles of the predicted structures and experimental patterns would give prominence to the mismatch in unit cell parameters and experimental profile parameters, such as describing peak shape, rather than providing a high-level comparison of the similarity of the simulated and experimental profiles. The predicted structure could be modified to use a unit cell extracted from the experimental data and the profile simulated using experimental parameters derived from Le Bail or Pawley fits of the experimental pattern. However, we feel that this could potentially give a false impression of the true comparability.

To rationalize this choice, we have now added the following text to the ESI (pg. 16):

“PXRD analysis for experimental and predicted TAPT.Cl, TT.Br and TTBT.Cl

Diffraction profiles were simulated for the structures most closely matching the experimental **TAPT.Cl**, **TT.Br** and **TTBT.Cl** patterns. Stack plots of the simulated and experimental patterns (Fig. 2g-i) show the similarity of the **TAPT.Cl** and **TT.Br** profiles in particular. Minor differences in the experimentally observed unit cell and predicted structure, and experimental effects such as background scattering and peak broadening tend to dominate the difference curves; hence these are not included. To investigate the differences in the **TTBT.Cl** profiles, constrained structural refinement of **TTBT.Cl** was performed as described below.”

c) The main text suggests that the tetrahedral tectons are not promising candidates for the formation of porous salts. I note, however, that the energy-density landscapes

shown in Extended Data Figure 1a reveals a spike of lower energy structures, some of which feature channels. Could such structures be accessed using various bulky template molecules,¹² which could subsequently be removed? Would this activated solid be stable, or would it collapse? Is it worthwhile pursuing such structures? I do not expect that this is addressed experimentally on this occasion, but it might be something that should be considered when discussing such landscapes.

Response: Yes, this is possible. Some of the porous structures predicted in the lower density region of the **TAPM.X** landscaped look potentially feasible (e.g., **TAPM_opt.chloridex4-QR-33-164790-3**, shown below), and several of these structures fall within a lattice energy range that might be accessible by solvent templating, which is at least 50 kJ mol⁻¹ above the global lattice energy minimum, based on our previous findings for porous structures of neutral molecules. We did not observe such structures here over a range of trialled solvents—and it is possible that the energy range for polymorphism is different for charged frameworks—but the idea of purposeful templating is an interesting idea (*c.f.*, *Nat. Chem.*, **2015**, 7, 153).

*Predicted structure **TAPM_opt.chloridex4-QR-33-164790-3***

We have not added an additional discussion of this for **TAPM.X** for reasons of manuscript length and because we gave a detailed analysis of alternative packings for the other molecules in Extended Data Figs. 4–7. However, taking the reviewer’s point about suggesting that tetrahedral tectons are unpromising, we have now softened this statement. This also reflects new preliminary data that we have, suggesting that the TAPM tecton might form porous structures when combined with other anions (pg. 6, lines 4–7).

Original text: “In both cases, the lowest energy structures were predicted to be dense and non-porous, suggesting that these salts were not promising candidates for CPOS materials, even though structurally analogous anionic tetrahedral sulfonates were used previously to create porous salts⁵.”

New text: “In both cases, the lowest energy structures were predicted to be dense and non-porous, suggesting that **TAPM** was not a promising candidate for **stable, permanently porous** CPOS materials, **at least with halide counterions**, even though structurally analogous anionic tetrahedral sulfonates were used previously to create porous salts⁵.”

d) Page 1, Line 16: The ‘metal-organic bonding’ is better described as ‘coordinate covalent bonding’.

Response: We have made this change.

e) Page 2, Lines 1-8: I would describe hydrogen-bonded organic frameworks (HOFs) as a type of extended framework structure. I agree that HOFs are held together by weaker interactions than those seen in MOFs and COFs, but despite this, HOFs are, in my opinion, still an example of an extended framework material.

Response: Agreed. We have tightened this sentence up a bit (new text in yellow).

“Porous crystalline solids can be divided into two classes: extended, covalently-bonded frameworks, such as metal-organic frameworks (MOFs)^{1,2} and covalent-organic frameworks (COFs)^{13,14}, and porous molecular crystals, such as hydrogen-bonded frameworks (HOFs)^{7,8,15,16} and porous organic cages (POCs)¹⁷.”

f) Page 7, Line 13: I suggest changing ‘14.3 × 8.5 Å’ into ‘14.3 Å × 8.5 Å’ or ‘14.3 × 8.5 Å²’

Response: We have changed this in three instances in this section.

g) Page 7, L32: How was the isostructurality quantified?^{13,14}

Response: The reviewer raises an interesting question. If one takes the definitions of isostructurality proposed in the first reference cited by the reviewer (ref. 13, Kálmán *et al.*, 1993)—that is, unit-cell similarity, atom substitution, etc.—then the term “isostructural” does not work for the **TT.Br** / **TTBT.Cl** pair. Indeed, to address exactly this problem, some of us have developed a new metric for isostructurality (*J. Am. Chem. Soc.*, **2022**, *144*, 9893). To introduce that metric here is beyond the scope of the revision, but it could constitute a follow-up paper. On reflection, we feel that ‘isostructural’ is not really the best choice of word here, and we have revised the sentence as follows, to use the word “isorecticular” (*i.e.*, “having the same or similar structural topology”).

Original text: “Powder X-ray diffraction data comparisons suggested that **TT.Br** and **TTBT.Cl** formed isostructural crystal packings, in agreement with the global energy minimum structures predicted by CSP (Figure 2h,i).”

New text (pg. 8, lines 29–32): “Comparison of experimental powder X-ray diffraction data with equivalent data derived from the global energy minimum CSP structures (Figure 2h,i) suggested that **TT.Br** and **TTBT.Cl** formed crystal packings that were broadly isorecticular: that is, extension of the organic linkers led to larger pores, as for isorecticular MOFs.”

We believe that the word “isorecticular” better captures the structural analogy for these two materials, acknowledging that the word “isostructural” can be problematic here.

We have also made the following changes, for the same reasons:

Page 2, line 6: “isostructural frameworks” → “structurally related frameworks”

Page 3, line 23: “isostructural” → “isoreticular” (also **pg. 3, line. 19; pg. 7, line 3; pg. 10, lines 17/18, pg. 12, line 10**).

We have retained the word “isostructural” on **pg. 6/line 10** and **pg. 6/line 23**, because here this does convey the correct meaning.

h) Page 8, Line 1: A part of the sentence seems to be missing: ‘as used to provide structural’.

Response: Thanks – this got truncated, it should have read “as used to provide structural information for other porous frameworks”. This is now corrected.

i) SI, Pages 7-8: The term ‘good’ solvent is well established in the community of solid-state chemists, but a broader readership might be better served with ‘effective solvent’.

Response: This was changed, as suggested.

j) SI, Page 25, Figure 18: The authors suggest that the changes in the PXRD of TTBT.Cl before and after iodine exposure are not significant. I note that the patterns are similar only up to about $7^\circ 2\theta$. Differences in the $7-50^\circ$ range are rather significant.

Response: The original text read as follows:

“We would expect significant changes in the PXRD pattern, given that 248 wt. % iodine is absorbed. However, we note that the original PXRD pattern is completely regenerated upon iodine removal (Figure 4d, main text) and that this can be done at least five times without any loss in iodine adsorption capacity (Figure 4c), demonstrating that these structural changes in the framework are fully reversible.”

This was meant to convey that we would expect significant changes in the PXRD, **and there are such changes**, but we can see now that this text was somewhat ambiguous. We have changed it to the following:

“We would expect that there could be significant structural changes to the **TTBT.Cl** framework and associated changes in the PXRD pattern, given that 248 wt. % iodine is absorbed. Moreover, even if the framework topology were unchanged, iodine is a good x-ray scatterer, and any ordered iodine in the framework pores might be expected to affect the PXRD pattern. Significant changes to the PXRD pattern are indeed observed after iodine sorption, but we note that the original PXRD pattern is completely regenerated upon iodine removal (Figure 4d, main text) and that this can be done at

least five times without any loss in iodine adsorption capacity (Figure 4c), demonstrating that any structural changes in the framework are fully reversible.”

k) SI document, Section 3: The alpha letter is capitalised. Please change to small caps. Also, various quantities and parameters (e.g., F and R_1) should be italicised.
Done.

Some numbers and symbols should be superscripted (e.g., mol⁻¹)
Checked throughout and changed where needed.

l) Dash symbols are used instead of the mathematical minus symbol throughout the text and in figures. Please correct this.
Checked throughout and changed where needed.

m) The units are inconsistently presented: kJ/mol and g cm⁻³. I suggest the use of kJ mol⁻¹ and g cm⁻³.
Checked throughout and changed where needed.

In conclusion, the study is a major milestone and truly innovative, and it is also exemplary in its execution. The authors have made an exceptional advance in the area of porous materials and materials science, and I look forward to seeing how the scientific community responds to their work.

Response: We thank the reviewer for these kind words and again for writing such a careful and detailed review. The suggestions have improved the clarity of the paper, particularly the question regarding quantification of isostructurality.

References

1. O. M. Yaghi, M. O’Keeffe, N. W. Ockwig, H. K. Chae, M. Eddaoudi, J. Kim, Reticular synthesis and the design of new materials. *Nature* 2003, 423, 705.
2. R. Freund, O. Zaremba, G. Arnauts, R. Ameloot, G. Skorupskii, M. Dincă, A. Bavykina, J. Gascon, A. Ejsmont, J. Goscianska, M. Kalmutzki, U. Lächelt, E. Ploetz, C. S. Diercks, S. Wuttke, The current status of MOF and COF applications. *Angew. Chem. Int. Ed.* 2021, 60, 23975.
3. D. Crawford, J. Casaban, R. Haydon, N. Giri, T. McNally, S. L. James, Synthesis by extrusion: continuous, large-scale preparation of MOFs using little or no solvent. *Chem. Sci.*, 2015, 6, 1645.
4. P. Guo, D. Dutta, A. G. Wong-Foy, D. W. Gidley and A. J. Matzger, Water sensitivity in Zn₄O-based MOFs is structure and history dependent, *J. Am. Chem. Soc.* 2015, 137, 2651.
5. C. Mottillo and T. Friščić, Carbon dioxide sensitivity of zeolitic imidazolate frameworks, *Angew. Chem. Int. Ed.*, 2014, 53, 7471.
6. F. Haase and B. V. Lotsch, Solving the COF trilemma: Towards crystalline, stable and functional covalent organic frameworks, *Chem. Soc. Rev.* 2020, 49, 8469.
7. N. Malek, T. Maris, M. Simard, J. D. Wuest, Molecular tectonics. Selective exchange of cations in porous anionic hydrogen-bonded networks built from derivatives of tetraphenylborate. *J. Am. Chem. Soc.* 2005, 127, 5910.

8. V. A. Russell, M. C. Etter, M. D. Ward, Layered materials by molecular design: structural enforcement by hydrogen bonding in guanidinium alkane- and arenesulfonates, *J. Am. Chem. Soc.*, 1994, 116, 1941.
9. M. K. Corpinot, D.-K. Bučar, A practical guide to the design of molecular crystals. *Cryst. Growth Des.* 2019, 19, 1426.
10. D. A. Bardwell et al. Towards crystal structure prediction of complex organic compounds - a report on the fifth blind test. *Acta Cryst. Sect. B*, 2011, 67, 535.
11. M. A. Reilly et al. Report on the sixth blind test of organic crystal structure prediction methods. *Acta Crystallogr. Sect. B*, 2016, 72, 439.
12. K. Travis Holman, Stephen M. Martin, Daniel P. Parker, and Michael D. Ward The generality of architectural isomerism in designer inclusion frameworks, *J. Am. Chem. Soc.* 2001, 123, 442.
13. A. Kálmán, L. Párkányi, G. Argay, Classification of the isostructurality of organic molecules in the crystalline state. *Acta Cryst. Sect. B*, 1993, 49, 1039.
14. P. Bombicz, What is isostructurality? Questions on the definition. *IUCrJ*, 2023, 11, 3.

Referee #2 (Remarks to the Author):

In this manuscript the Authors presented the first time application of crystal structure prediction (CSP) to crystalline porous organic salts (CPOS). In these materials salt anions take the role of nodes, connected by positively charged organic linkers, a charge reversal compared to the nodes and linkers found in MOFs.

The key novelty of this work is the demonstration that CSP can reliably predict the structures of CPOS materials. The success of the method was tested on four linker types in combination with two halide anions, resulting in eight predicted energy landscapes. In all cases, the experimentally-observed structures were matched against the low energy predicted structures. Clearly CSP is a powerful and robust method for designing such materials computationally. Nonetheless, I would like to point at several aspects of the study, which, in my opinion need to be improved:

1) The key difference between MOFs and CPOS is not only the reversal of charges on nodes and linkers, but a different mode of binding between nodes and linkers: covalent coordination bonds in MOFs and non-covalent ionic interactions in CPOS. For one thing, this leads to fundamental differences in the CSP approaches needed for both of these types of materials, which would be useful to discuss in the manuscript.

Response: This is correct; the technical CSP requirements here are quite different to those for structure prediction in MOFs. There are, as yet, very few papers on *a priori* CSP for MOFs involving lattice energy calculations, and we cited the key papers (now Refs. 23, 24) in the initial submission (pg. 2, line 20), as follows:

“The area of MOFs has also seen recent developments in CSP^{23,24} that could be used to anticipate likely stable structures for particular metal-linker combinations.”

We have now expanded this text to address the reviewer’s point (pg. 2, lines 21–27):

“The area of MOFs has also seen recent developments in CSP^{23,24} that could be used to anticipate likely stable structures for particular metal-linker combinations; the covalent bonding between metal nodes and organic linkers requires periodic density functional theory to adequately describe the relative energies of alternative structures, unlike organic molecular CSP where intermolecular force fields can often capture the balance between competing non-bonded interactions. This makes structure-searching much more expensive in MOF CSP and, so, these studies have made heavy use of symmetry to guide the placement of MOF building blocks during random structure searching to reduce computational expense.”

From a material stability point of view, it would be interesting to quantify and compare the energies of these ionic interactions with the typical coordination bonds found in MOFs.

Response: It is difficult to generalize too much about the strength of ionic bonding and coordinate covalent bonding because they both span quite a large and rather continuous energy range, depending on the chemical functionality (see figure below). However, broadly speaking, coordinate covalent bonding can be both stronger and more directional than ionic bonding.

Figure taken from: *Cryst. Growth Des.* **19**, 1426 (2019)

We expressed this in the introduction to the paper (pg. 2, lines 4–7) as follows:

“Porous frameworks exploit strong, directional covalent or coordinate covalent bonding, which underpins the isorecticular principle², whereby series of structurally related frameworks can be synthesized. By contrast, porous molecular crystals involve weaker, non-covalent intermolecular interactions.”

To re-emphasize this, and to address the Reviewer’s point, we now return to this again in the outlook (pg. 14, lines 1–3, new text highlighted):

“We see CSP as the key to exploring this area because the ionic bonding in these salts is weaker and less directional than for most MOFs, and CSP allows us to evaluate the propensity for new combinations of organic cations and counterions to form stable, porous crystals prior to synthesis.”

The discussion of bonding directionality has now also been expanded in response to helpful comments raised by Reviewer 3, below, to point out that some ionic bonding is more directional than others (pg. 13, line 27 *et seq.*).

How does that reflect on the thermal stability of CPOS, as well as mechanical properties of these materials?

Response: The thermal stability of the materials after desolvation were tested by TGA (Supporting Figs. 12–14 in the initial submission). The stabilities are modest, with decomposition occurring between 150–200 °C for the three porous systems **TAPT.Cl**, **TT.Br** and **TTBP.Cl**. However, these materials show good stability and recyclability for applications such as iodine capture and release below these decomposition temperatures (Fig. 4c,d; Supporting Figs. 20–22).

In response to comments made by Reviewer 1, we have now also investigated the *water* stability of the materials, and there is a new section on that (pg. 12, line 28 *et seq.*, Supporting Figs. 23 & 24; in *précis*, it varies considerably depending on the hydrophobicity of the organic linker).

We do not have ready access to mechanical testing facilities, but we agree that this could be an interesting follow up study. Anecdotally, these materials form relatively brittle crystals, and we would not expect the mechanical stability to be exceptionally good (or bad) in comparison with comparably porous MOFs, but to say anything more useful here would require measurements.

2) The synthesized materials demonstrate favorable iodine sorption performance. However, it is recognized in the manuscript that calculation of pore volume in the predicted structures is not a quantitatively-accurate predictor for sorption capacity. Perhaps some GC-MC/MD simulations, or some alternative method, could be applied, at least for selected structures, to show whether the iodine sorption capacity can be predicted quantitatively.

Response: This is a good suggestion but the simulation of this, particularly if one allows for framework flexibility, which is probably needed here, is beyond the requested timescale for these revisions. It is something that we will follow up in future work.

3) I would like to ask to replace PXRD comparison figures, where the pattern simulated from the predicted structure is shown above the experimental pattern, with Rietveld refinement plots. Showing the agreement between the experimental and calculated (refined) profile, together with the difference curve would be a stronger proof for the agreement between predicted and synthesized structures.

This is particularly necessary for the structure of TTBT.Cl, where simulated and experimental peak intensities are widely different. Rietveld refinement could clarify, whether this is caused by preferred orientation, presence of guest molecules in the pores or issues with the geometry of the structure itself.

The description of the PXRD analysis of TTBT.Cl got me somewhat confused. The Authors report indexing the powder pattern with a monoclinic C-centered cell: $a=27.581$, $b=47.314$, $c=8.0051$, $\beta = 97.61$, saying that this cell is similar to that of the lowest energy predicted structure. But looking at the CIF of the predicted structure I found a cell with dimensions: $a=80.60$, $b=40.11$, $c=6.16$, $\beta=90.06$, which is very different. This needs to be clarified.

Response: The reviewer is referring here to the CSP structure **TTBT.Cl/1**, which has a high predicted lattice energy (see Extended Data Fig. 6). The relevant cif file for the global minimum predicted structure is **TTBT.Cl/5**. This does have cell parameters that are close the refined experimental structure, as follows:

	TTBT.Cl (expt refined)	TTBT.Cl/5 (CSP)
Space Group	C2/c	C2/c
a [Å]	27.5587(3)	27.9505
b [Å]	47.3544(5)	47.2742
c [Å]	7.9513(9)	8.1603
α [°]	90	90.0
β [°]	97.53(1)	97.872
γ [°]	90	90.0

If we display these two structures side by side, then the structural similarity is clear (this has now been included in Extended Data Fig. 3d):

Extended Data Fig. 3. ... d, Comparison of refined experimental structure for TTBT.Cl and global minimum CSP structure, TTBT.Cl/5.

From the perspective of porosity, at least, these two structures can be considered as equivalent.

With regards to the reviewer's request for difference curves to be added to Fig. 2g-i, please see the previous response to Reviewer 1. The patterns/structures for **TAPT.Cl** and **TT.Br** were not refined, therefore difference curves for the profiles simulated from the predicted structures and experimental patterns primarily indicate differences in how the diffraction experiment is modelled in the simulation, and small differences in the unit cells.

The experimental PXRD for **TTBT.Cl** was indeed used as the basis for structural refinement of the predicted model as described in the ESI. We prefer the term "structural refinement" here over "Rietveld refinement" due to the need for heavily constraints on the structural model as, arguably, the latter term would imply refinement of all the atomic positions. We also acknowledge that some practitioners of PXRD analysis may indeed find "structural refinement" an overstatement.

4) Related to the previous point, **TTBT.Cl** is clearly the most challenging system of all presented. The linker is conformationally flexible, there is a possibility of the experimental structure having a different space group symmetry from the predicted model. I think some of these ambiguities could be resolved by a more detailed analysis of the experimental PXRD pattern, starting from the model of the predicted structure, rather than attempting full *ab initio* structure solution. Whilst refinement of all atom positions is not feasible, given the low data quality, perhaps molecular fragments defined via z-matrix with some flexible torsion angles could be used?

Response: We appreciate the reviewer's suggestion of using the predicted structure directly as a starting model for Rietveld refinement against the experimental PXRD. This would fundamentally assume that the experimentally observed structure is the same as (or is very close to) the predicted structure used as the initial model. This would also include the unit cell from the prediction, and would not provide additional information on the true symmetry. Indeed, the possibility of higher space group symmetry was only identified because the PXRD was analysed independently from the predicted structure.

The approach of using an experimentally determined set of lattice parameters, followed by an *ab initio* structure solution and constrained structural refinement was used to allow for the structural analysis to be led by the experimental data as far as possible, while supplementing the low resolution PXRD with information available from the predicted structure. In principle, this allows for a different structure that is more consistent with the experimental data be identified during the structure solution. Including the *ab initio* step also enabled reduced bias in the modelling of solvent/hydrate positions and occupancies. It should be noted that the structure solution used the experimental unit cell selected based on its similarity to the global minimum predicted structure and the conformation from the predicted structure. Hence, as the *ab initio* solution had a packing arrangement consistent with the prediction, the two approaches were largely analogous.

In terms of further modelling of the molecular conformation, we have chosen a conservative approach due to the relatively low real-space resolution of the diffraction data. The broadening of the diffraction peaks and presence of high and structured background in the experimental PXRD of **TTBT.Cl** indicates the presence of significant disorder in the crystalline solid, as discussed in the ESI. This is also evident in the electron diffraction images (e.g., Extended Data Fig. 3b). The PXRD has few discernible diffraction peaks at $2\theta \geq 24^\circ$, which corresponds to a real-space resolution of approximately 3.7 Å. Although it would be possible to include additional parameters for molecular flexibility in the refinement model, we feel that because of the limited information present in the diffraction pattern at shorter length scales, it would be difficult to justify drawing conclusions about the conformation based on the result.

We have added the following text to further explain this (pg. 9, lines 10–12):

“The halos observed in the FFT images suggest that there is uncorrelated disorder in these two materials, particularly for **TTBT.Cl** (Extended Data Fig. 3b).”

We have also modified the ESI to explicitly include the possibility of disordered molecular conformation contributing to the broadened diffraction, as follows:

“The pattern has relatively low resolution with significant broadening of the diffraction peaks, probably due to a combination of the short ordered length scale, the presence of disordered residual solvent and molecular conformation.”

To summarize, I enjoyed reading the manuscript and I believe that this work represents a major milestone for the design of porous materials. The article is written well, the methodology and the results are clearly explained.

Referee #3 (Remarks to the Author):

Review of “Porous isorecticular non-metal organic frameworks”
O’Shaughnessy, et al.

This manuscript by O’Shaughnessy, et al. describes a combined computational and experimental approach to the synthesis of porous organic salts, analogous to MOFs but flipping the script with respect to the charge on the node (negatively charged halide ions) and linkers (positively charged polyvalent ammonium ions). I’ve had the opportunity to review many MOF articles in which computations, either data mining or ab initio methods, are used to predict new structures and sometimes their adsorption properties. The O’Shaughnessy submission, however, is a rare example of computations confirmed by experiment. Here, CSP is used to identify the lowest energy polymorphs and then the authors actually crystallized the lowest energy forms for two of the linkers and one of the lowest for a third (i.e., near the low energy” tip of the CSP grouping. This is unique, and in my view will be of interest to the broader community, particularly those interested in de novo design of organic solids as well as those interested in using CSP reliably to identify polymorphs. The authors posit sound reasoning that explains the relationship between packing density and the coordination

number of the linker groups about the spherical anionic nodes. I would have been supportive of publication if the manuscript ended here, but the authors also demonstrated iodine adsorption, which is reversible with respect to regenerating the original host structure. This is a phenomena that seems deserving of a more detailed investigation, but it is sufficient for this submission as it demonstrates clearly that using CSP to accelerate synthesis of a functional material, here in a relatively unexplored class of materials, can have a useful application (here new materials for radioactive iodine capture). I strongly support publication. The conclusions are well-supported by the data, and the manuscript is well-written with clarity and a logical flow. I have some questions/comments, however, offered here in the spirit of improving clarity.

(1) A comment about the term “HOF”. There is a lack of clarity and a lot of confusion about this term, because it was first coined by Banglin Chen to include hydrogen-bonded organic frameworks that were porous. This led to labeling them as HOF-1, HOF-2, etc., I believe in Y2013. But of course hydrogen-bonded organic frameworks were known decades before. I like that the authors have largely used the POS alternative, although this would not apply to neutral compounds.

Response: Agreed. To make this clear, where we first introduce the term ‘HOF’ (pg. 2, line 4) we have now added a reference to an early paper by Wuest and colleagues (new reference 16), illustrating that this field did not start in 2013. We could have chosen other studies from that period, but we feel this paper espoused many of the key concepts that have been developed subsequently by others for neutral frameworks.

16. Simard, M., Su, D. & Wuest, J. D., Use of hydrogen-bonds to control molecular aggregation - self-assembly of 3-dimensional networks with large chambers, *J. Am. Chem. Soc.*, **113**, 4696–4698 (1991).

(2) Regarding point (1), in my view, claims of porosity in HOFs is tenuous in many examples. The authors correctly note that many HOFs are not stable. One example that is irrefutable is Brekalo, et al. (*Angew. Chem. Int. Ed.* 2020, 59, 1997–2002, DOI: 10.1002/anie.201911861), which describe guanidinium benzenedisulfonate, which is truly porous and was distinguished from HOFs in general using the term p-HOF. The authors may want to cite Brekalo, et al. in the introduction among their example POS materials.

Response: This paper is relevant on two levels because the material was shown to be porous and, in the long term, metastable. We have now added a reference to it (new ref. 29) in the introduction part as follows (pg. 3, lines 1–4):

“Brekalo *et al.* showed that guanidinium organodisulfonates, while formally metastable with respect to dense packings, can retain microporosity for extended periods.²⁹”

We also now refer to this study again where we discuss the CSP calculations (pg. 6, lines 32–34):

“This provides an insight that organic salts might be more suitable for creating intrinsic porosity, which is important for applications because metastable crystals are subject to porosity loss by densification.²⁹”

(3) At the risk of revealing my identity (in any case I signed my review below), I would like to note that polymorphism has never been observed among the more than 700 crystalline guanidinium organosulfonate compounds (either guest-filled or guest free). These compounds are salts. Some of these compounds have polyvalent sulfonate “linkers” connected to guanidinium ions through hydrogen bonding, some with crystal structures have topologies similar to those described by O’Shaughnessy, although densely packed because of guest occupation or interdigitation of organosulfonate residues. But can the absence of polymorphism in guanidinium organosulfonates be explained by the directional nature of the hydrogen bonds between guanidinium ions and sulfonate groups? Conversely, would negatively charged sulfonates linked to spherical cations be more likely to exhibit polymorphism because of the absence of directional bonding? I have never considered the link between directional hydrogen bonding and the absence of polymorphism, and I would be very interested to know if the authors have considered it. If so, maybe worth mentioning in the article?

Response: This is an interesting suggestion and very plausible – it is something that we will explore in the future. Indeed, we have already started to explore ammonium sulfate salts, which have anions that are less symmetrical than halides. From the CSP point of view, it is currently difficult to predict propensity for polymorphism because methods almost always overpredict polymorphism. However, methods are being developed to address overprediction in CSP, using enhanced sampling molecular dynamics (*Cryst. Growth Des.* 2020, 20, 10, 6847–6862) or Monte Carlo sampling (*Proc. Natl. Acad. Sci. U.S.A.*, 2023, 120 (23) e2300516120). While it has not yet been possible to apply such methods here because of the complexity of these salt systems, and the huge numbers of structures on their CSP landscapes, development of these methods should soon allow us to address these questions computationally, to complement experimental screening.

For now, we have added the following text to the outlook section of the paper (pg. 13, lines 27 *et seq.*)

“We observed polymorphism for **TAPM.X** salts (Extended Data Fig. 1), and CSP calculations suggest that other ammonium halide salts might in principle be polymorphic, too (Extended Data Figs. 4–6), although CSP is known to overpredict polymorphism^{49,50}. Interestingly, polymorphism has not been observed in crystalline guanidinium organosulfonate materials^{3,4,29}. This might be due to more directional hydrogen bonding between guanidinium ions and sulfonate groups restricting the possibilities for low energy crystal packings, in comparison with ammonium halide salts that comprise simple spherical anions. This could have broader implications for the design of non-metal organic frameworks using anions other than halides.”

(4) Page 7, line 6: “Figure 2a-c. In all three cases, these matches were found to lie at the tip of a ‘spike’ in the CSP landscape, and for TT.Br and TTBT.Cl these structures

corresponded to the predicted global “ energy minimum structures. Again, this is a remarkable correspondence of CSP and experimental outcome.

Response: Yes – it was great to see this when we first synthesized the materials because while they satisfied our simple design hypothesis (charge adjacency), there are multiple crystal packings that can achieve this. For example, for **TTBT.Cl**, essentially all the low energy predicted structures have four close $\text{NH}_3^+/\text{Cl}^-$ neighbours (Figure 3e). In a sense, the strategy is ‘pseudo-isorecticular’: that is, there are some underlying structural rules, but these rules are not trivially intuitive as for, say, the BCC packing in ammonium chloride.

(5) Page 7, line 9: Maybe add some clarity to this statement: “These three crystal packings were all isostructural and comprised two distinct one-dimensional (1-D) pore channels, as labelled in Figure 2d-f.” At this stage of the manuscript, crystal packings in the experimental structures were deduced from CSP and comparison to PXRD, not single crystal X-ray diffraction (later, single crystal X-ray diffraction confirmed one of the structures).

Response: This is a good point; we are talking here about the CSP structures. To make this clearer, we have modified the relevant paragraph as follows (pg. 8, line 3 *et seq.*; new text is highlighted). Note that the word change from “isostructural” to “isorecticular” addresses a point made by Reviewer 1, above:

“The **three predicted crystal packings that best matched the experimental data** were all **isorecticular** and comprised two distinct one-dimensional (1-D) pore channels, as labelled in Figure 2d-f. The first pore (A) is defined by clusters or ‘tubes’ of the protonated amines and the halide counterions; it is cylindrical, highly charged, and has a narrow pore diameter (4.86–5.87 Å). The second pore (B) is roughly diamond in shape, less polar, and defined by the aromatic linkers; this pore diameter is significantly larger in **TTBT.Cl** (14.3 Å × 8.5 Å) than in **TAPT.Cl** (7.9 Å × 4.6 Å) or **TT.Br** (7.0 Å × 5.8 Å), while the dimensions of the ionic pore (A) are the same in all three **predicted** structures. This dual channel structure leads to **predicted** pore volumes in these trigonal amine salts that are higher than for 4,4',4'',4'''-(ethene-1,1,2,2-tetrayl)tetraaniline (ETTA) salts³⁴.”

(6) The “partial solvate” in TAPT.Cl is a bit indefinite. Can the authors state the amount of solvent, and the true porosity of the single crystal when solvent is accounted for?

Response: The refined composition for the **TAPT.Cl** single crystal structure was given in the Supporting Information (pg. 11) as follows:

Formula $3(\text{Cl})$, $\text{C}_{24} \text{H}_{24} \text{N}_3$, $1.25[\text{C}_6\text{H}_5\text{Cl}]$, $1.5[\text{H}_2\text{O}]$

We now also give this information in the main text (pg. 8, line 19).

These solvent guests occupy 78% of the void volume, but this is slightly notional with regard to porosity because sorption measurements apply a high dynamic vacuum, and guests would be removed under such measurement conditions. As such, we chose not

to discuss the void volume in this solvate as porosity, but we have instead now added the following sentence to clarify the solvent occupancy (pg. 8, line 20):

“In this solvate, the solvent occupies around 78% of the void volume.”

We also note that in addressing this comment, we recalculated all the pore volumes using Olex2, and a description of the methodology is now given in the ESI (pg. 3). This gives slightly different values to the ones (derived from Materials Studio), but the overall conclusions and the ordering of porosity for the materials remains the same (pg. 8, lines 11–13).

(7) Regarding iodine adsorption, it may be a good idea to express the amount of adsorption as mole iodine/mol host, as mol/g may not be so important for the application stated here, and mol/mol is more helpful to understanding the occupancy of iodine in the host.

Response: Most of the iodine capture literature quotes wt. % values (e.g., see recent review, Ref. 43), but we agree that mol/mol is more useful for the supramolecular community. We have amended the text to quote both, as follows:

“The material with the largest predicted pore volume, **TTBT.Cl**, showed the highest iodine uptake (248 wt. % / 6.8 mol/mol). The isoreticular salts **TT.Br** and **TAPT.Cl** showed uptakes of 213 wt. % (4.99 mol/mol) and 211 wt. % (3.83 mol/mol), respectively.”

Reviewer Reports on the First Revision:

Referees' comments:

Referee #1 (Remarks to the Author):

The authors have carefully addressed all comments in full. They have provided clarifications where requested, and presented new results, without disrupting the thought-provoking and engaging narrative. The revisions have clearly enhanced an already exceptional manuscript.

Referee #2 (Remarks to the Author):

In my opinion, the revised version of the manuscript addresses all the comments, and I have no further questions to ask.

At this point I am happy to recommend this article for publication, and would like to thank the Authors for their excellent work.

Referee #3 (Remarks to the Author):

The authors have addressed my comments and in my view, the manuscript certainly is worthy of publication. I took the opportunity to review the responses to the other reviewers as well, and the authors have addressed these as well. I expect the article to have substantial impact. It was a quite enjoyable read and I look forward to seeing it published.